# Organelle proteomic profiling reveals lysosomal heterogeneity in association with longevity

Yong Yu[1,2]*[†], Shihong M Gao[3,4†], Youchen Guan[4,5†], Pei-Wen Hu[2], Qinghao Zhang[2], Jiaming Liu[1], Bentian Jing[1], Qian Zhao[4], David M Sabatini[6], Monther Abu-Remaileh[7,8], Sung Yun Jung[9], Meng C Wang[2,4]*

[1]State Key Laboratory of Cellular Stress Biology, School of Life Sciences, Faculty of Medicine and Life Sciences, Xiamen University, Xiamen, China; [2]Huffington Center on Aging, Baylor College of Medicine, Houston, United States; [3]Developmental Biology Graduate Program, Baylor College of Medicine, Houston, United States; [4]Janelia Research Campus, Howard Hughes Medical Institute, Ashburn, United States; [5]Molecular and Cellular Biology Graduate Program, Baylor College of Medicine, Houston, United States; [6]Institute of Organic Chemistry and Biochemistry, Prague, Czech Republic; [7]Institute for Chemistry, Engineering and Medicine for Human Health (ChEM-H), Stanford University, Stanford, United States; [8]Department of Chemical Engineering and Genetics, Stanford University, Stanford, United States; [9]Department of Molecular and Cellular Biology, Baylor College of Medicine, Houston, United States

*For correspondence:
yuy@xmu.edu.cn (YY);
mengwang@janelia.hhmi.org
(MCW)

†These authors contributed
equally to this work

Reviewing Editor: Kang Shen,
Stanford University, United
States

**Abstract** Lysosomes are active sites to integrate cellular metabolism and signal transduction. A collection of proteins associated with the lysosome mediate these metabolic and signaling functions. Both lysosomal metabolism and lysosomal signaling have been linked to longevity regulation; however, how lysosomes adjust their protein composition to accommodate this regulation remains unclear. Using deep proteomic profiling, we systemically profiled lysosome-associated proteins linked with four different longevity mechanisms. We discovered the lysosomal recruitment of AMP-activated protein kinase and nucleoporin proteins and their requirements for longevity in response to increased lysosomal lipolysis. Through comparative proteomic analyses of lysosomes from different tissues and labeled with different markers, we further elucidated lysosomal heterogeneity across tissues as well as the increased enrichment of the Ragulator complex on Cystinosin-positive lysosomes. Together, this work uncovers lysosomal proteome heterogeneity across multiple scales and provides resources for understanding the contribution of lysosomal protein dynamics to signal transduction, organelle crosstalk, and organism longevity.

## Editor's evaluation

The authors present a powerful tool to unbiasedly identify lysosome-associated proteins in *C. elegans*, and they provide a compelling, in-depth assessment of how this method can be used to understand longevity pathways and identify novel proteins. Understanding lysosomal differences in specific tissues or in response to different longevity conditions are exciting as it provides new insight into how organelles could control specific homeostasis responses. This valuable tool and proteomics datasets also represent a great resource for the *C. elegans* community and should pry open new studies on the regulation and role of the lysosome at the organismal level.

## Introduction

Lysosomes are membrane-bound organelles specialized to constitute an acidic environment in the cytosol. Lysosomes carry many proteins that are essential for maintaining lysosomal activities and mediating lysosomal regulatory effects. Inside the lysosomal lumen, a series of acidic hydrolases, including lipases, proteases, glucosidases, acid phosphatases, nuclease, and sulfatases, are responsible for the degradation and recycling of extra- and intracellular materials delivered through endocytic, phagocytotic, and autophagic processes (*Appelqvist et al., 2013*; *Ballabio and Bonifacino, 2020*; *Lawrence and Zoncu, 2019*). Additionally, on the lysosomal membrane, a group of integral transmembrane proteins play crucial roles in the maintenance of luminal acidic pH and ion homeostasis, the control of lysosomal membrane potential and export of metabolic products, as well as the regulation of organelle interaction and signal transduction (*Ballabio and Bonifacino, 2020*; *Lawrence and Zoncu, 2019*). For example, the lysosomal vacuolar-type $H^+$-ATPase (v-ATPase) on the membrane is the primary driver for the active accumulation of protons in the lysosomal lumen, which also requires a neutralizing ion movement mediated by ion channels and transporters (*Graves et al., 2008*; *Nicoli et al., 2019*). In addition, v-ATPase coordinates with lysosomal amino acid transporter SLC38A9 and lysosomal cholesterol exporter NPC1 in regulating the activation of mechanistic/mammalian target of rapamycin complex I (mTORC1) by amino acid and lipid cues (*Castellano et al., 2017*; *Wang et al., 2015*). The recruitment of mTORC1 to the lysosome is mediated by RagA/B and RagC/D GTPase heterodimers that are associated with the scaffold protein complex Ragulator tethered on the lysosomal membrane (*de Araujo et al., 2017*). Through interacting with Axin, Ragulator also mediates the activation of AMP-activated protein kinase (AMPK) on the lysosomal surface (*Zhang et al., 2014*). Furthermore, lysosomes are not static, isolated organelles, instead they are highly mobile vesicles that undergo frequent movements in both anterograde (nucleus-to-periphery) and retrograde (periphery-to-nucleus) directions and form dynamic interactions with other organelles including endosomes, autophagosomes, endoplasmic reticulum (ER), and mitochondria (*Ballabio and Bonifacino, 2020*; *Pu et al., 2016*). These trafficking and interaction processes are mediated by lysosomal integral transmembrane proteins as well as diverse proteins that are recruited to lysosomes in response to different extra- and intracellular inputs (*Ballabio and Bonifacino, 2020*; *Pu et al., 2016*).

Lysosomes control numerous cellular processes, and dysfunction of lysosomes has been linked with various diseases, such as lysosomal storage disorders (*Ballabio and Gieselmann, 2009*; *Platt et al., 2012*), Alzheimer's disease (*Nixon and Cataldo, 2006*), Parkinson disease (*Navarro-Romero et al., 2020*), and some types of cancer (*Davidson and Vander Heiden, 2017*; *Fehrenbacher and Jäättelä, 2005*). Emerging evidence also suggests that lysosome functions as a central regulator of organism longevity, through its involvement in autophagy and its modulation of metabolic signaling pathways. The induction of autophagic flux has been observed in multiple pro-longevity states, and is required for the pro-longevity effects caused by those genetic, dietary, and pharmacological interventions, such as reduced insulin/IGF-1 signaling, caloric restriction, and spermidine treatment (*Hansen et al., 2018*; *O'Rourke et al., 2013*; *Lapierre et al., 2011*). On the other hand, lysosomes are now recognized as the key platform to modulate the activities of mTORC1 and AMPK signaling, two well-characterized longevity regulating pathways (*Savini et al., 2019*). In addition, our studies have discovered lysosomal lipid messenger pathways that are induced by a lysosomal acid lipase LIPL-4 and promote longevity via both cell-autonomous and cell-nonautonomous signaling mechanisms (*Folick et al., 2015*; *Ramachandran et al., 2019*; *Savini et al., 2022*; *Wang et al., 2008*). Given the importance of lysosomes in regulating longevity, it will be crucial to understand how changes in the lysosomal protein composition are associated with longevity regulation.

To systemically profile the protein composition of lysosomes, methods have been developed to purify lysosomes using gradient centrifugation (*Gao et al., 2017*; *Lübke et al., 2009*; *Markmann et al., 2017*; *Schröder et al., 2010*). More recently, a lysosome immunoprecipitation method, which uses anti-HA (human influenza virus hemagglutinin) antibody-conjugated magnetic beads to immunopurify lysosomes from mammalian cells expressing transmembrane protein 192 (TMEM192) fused with three tandem HA (3×HA) epitopes, has further improved the specificity and speed of lysosomal isolation (*Abu-Remaileh et al., 2017*). This rapid isolation method has facilitated follow-up mass spectrometry (MS)-based proteomics as well as metabolomics analyses (*Abu-Remaileh et al., 2017*; *Eapen et al., 2021*; *Laqtom et al., 2022*).

In the present study, we have applied an immunoprecipitation-based method for rapid isolation of lysosomes from live adult *Caenorhabditis elegans* using transgenic strains expressing lysosomal membrane proteins tagged with 3×HA (Lyso-Tag). We then conducted large-scale proteomic profiling using isolated lysosomes and remaining non-lysosomal fractions, to determine the enrichment of each identified protein on the lysosome. Based on these analyses, we have defined a lysosome-enriched proteome and compared it between wild-type (WT) and long-lived worms, revealing lysosomal protein composition changes associated with longevity. We have also generated transgenic strains expressing Lyso-Tag specifically in four major somatic tissues, the hypodermis (epidermis), muscle, intestine (digestive tract/fat tissue/liver), and neurons, leading to the discovery of lysosomal proteome heterogeneity in different tissues. Furthermore, by comparing the lysosome-enriched proteome with LAMP1/LMP-1 Lyso-Tag and the one with Cystinosin/CTNS-1 Lyso-Tag, we discovered that the Ragulator complex and other mTORC1 regulators exhibit increased enrichments on lysosomes containing the cysteine transporter Cystinosin.

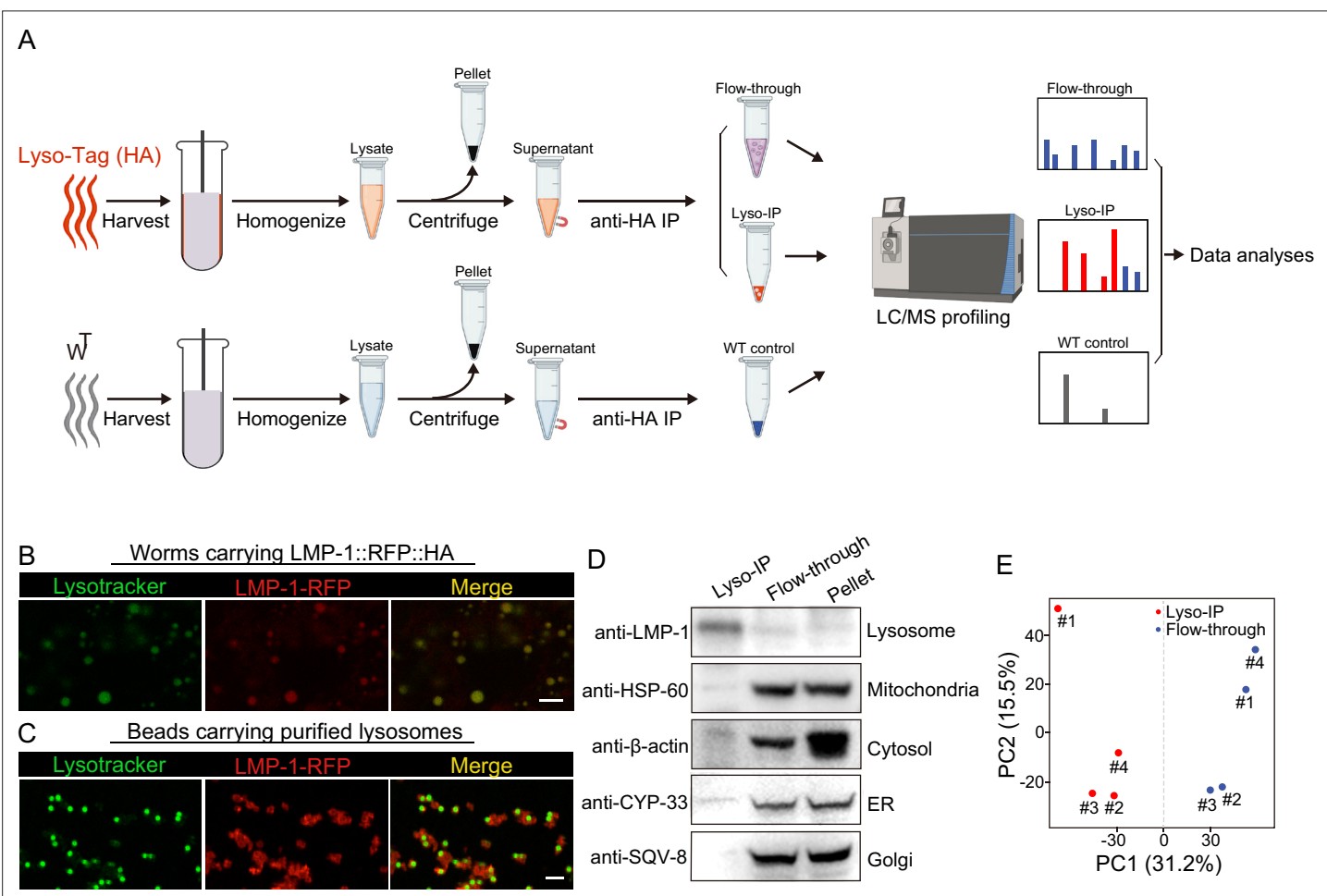

**Figure 1.** Rapid lysosome isolation coupled with proteomic profiling. (**A**) Schematic of the workflow for immunoprecipitation-based lysosome purification (Lyso-IP) and mass spectrometry-based proteomic profiling to identify lysosome-enriched proteomes in *C. elegans*.(**B**) Example images of transgenic strains carrying LMP-1 Lyso-Tag (LMP-1::RFP-3×HA) with LysoTracker staining to mark lysosomes in vivo. Scale bar = 5 μm. (**C**) Example images of beads carrying purified lysosomes from Lyso-IP with LysoTracker staining to mark intact lysosomes in vitro. Scale bar = 5 μm.(**D**) Western blot for protein markers of different subcellular compartments using purified lysosomes (Lyso-IP), paired non-lysosomal fractions (Flow-through) or Pellet. (**E**) Principal components analysis (PCA) of four independent biological replicates of Lyso-IP and Flow-through samples.

The online version of this article includes the following source data and figure supplement(s) for figure 1:

**Source data 1.** Western blots shown in *Figure 1*.

**Figure supplement 1.** Analysis of *LysoTg* lines and Lyso-IP profiling in wild-type (WT) worms.

# Results

## Map lysosome-enriched proteome systemically in *C. elegans*

To comprehensively reveal proteins that are enriched at the lysosome, we have applied rapid lysosome immunoprecipitation followed by MS-based proteomic profiling (Lyso-IP) (*Figure 1A*). We first generated a transgenic strain overexpressing the lysosome-associated membrane protein, LMP-1 (*Eskelinen, 2006*) fused to both 3×HA and RFP (LMP-1 *LysoTg*) under the whole-body *sur-5* promoter. Fluorescence imaging of RFP confirmed the lysosomal localization of the LMP-1 fusion protein in live organisms and made it possible to follow purified lysosomes in vitro (*Figure 1B*). The presence of transgenes does not affect worms' developmental timing and lifespan (*Figure 1—figure supplement 1A, B*). The 3×HA epitope tag is used to purify lysosomes from homogenized worm lysate via immunoprecipitation using anti-HA antibody-conjugated magnetic beads (*Figure 1A*). In general, about 160,000 worms at day-1 adulthood were harvested and homogenized. Upon centrifugation to remove debris and nuclei, 3×HA-tagged lysosomes were immunoprecipitated and separated from other cellular content (flow-through controls, *Figure 1A*). The whole process from harvesting worms to purified lysosomes takes around 25 min. Many purified lysosomes were able to take up LysoTracker probes and exhibit positive fluorescence signals, indicating that they remain intact with an acidic pH, while there are also some broken lysosomes losing LysoTracker staining (*Figure 1C*). When blotting with antibodies against different organelle markers, we found that the purified lysosomes show no or nearly no protein markers of other organelles, including HSP-60 (mitochondria heat shock protein) (*Hartl et al., 1992*; *Mayer, 2010*), CYP-33E1 (ER cytochrome P450) (*Brown and Black, 1989*), SQV-8 (Golgi glucuronosyltransferase) (*Hadwiger et al., 2010*), and β-actin (cytoskeleton) (*Figure 1D*), while the flow-through controls show these protein markers but nearly no lysosomal protein marker LMP-1 (*Figure 1D*). Together, these results demonstrate the efficacy of the Lyso-IP approach to enrich lysosomal proteins.

Next, we conducted proteomic profiling of purified lysosomes with their paired flow-through controls (*Figure 1A*). The correlation analysis shows good reproducibility among four independent biological replicates (*Figure 1—figure supplement 1C*), and the principal components analysis (PCA) shows a clear separation between Lyso-IP replicates and flow-through controls (*Figure 1E*). In parallel, we also conducted immunoprecipitation using homogenized lysate from WT worms that do not carry a Lyso-Tag and then analyzed proteomic profiles of three independent samples as non-tag controls (*Figure 1A*).

Based on these proteomic data, we used three criteria to define lysosome-enriched proteins: first, their levels in the purified lysosomes are 10-fold or higher than those in the flow-through controls (*Figure 2A*); second, their enrichments can be repeated in all biological replicates (*Figure 2A*); and lastly, their enrichments over non-tag controls are more than 2-fold (*Figure 2B*). Together, 216 lysosome-enriched candidates were identified from more than 6000 detected proteins, and 178 candidates have mammalian homologs (*Figure 2—figure supplement 1*, *Supplementary file 1*). This lysosome-enriched proteome consists of 83 membrane transporters and channels, 47 enzymes, 26 signaling factors, 12 structural components, and 6 involved in vesicle trafficking (*Figure 2C*). These include known lysosomal proteins, such as various lysosomal Cathepsins that catalyze protein degradation (*Turk et al., 2012*), lysosomal specific ARL8 GTPase that mediates lysosome trafficking (*Nakae et al., 2010*), and subunits of lysosomal v-ATPase that pumps protons into the lysosomal lumen to maintain an acidic pH (*Forgac, 2007*; *Supplementary file 1*).

Lysosomal v-ATPase consists of both V0 and V1 domains that are associated with the lysosomal membrane and face the cytosol, respectively (*Figure 2D*). Reversible dissociation of the V1 and V0 domains responds to nutritional signals and plays a crucial role in the regulation of the lysosomal v-ATPase activity (*Kane, 1995*; *McGuire and Forgac, 2018*; *Ratto et al., 2022*; *Stransky and Forgac, 2015*). Except for VHA-18 (V1 H subunit), we were able to detect all other subunits of lysosomal v-ATPase, including VHA-5, 6, 7 and UNC-32 (V0 a subunits), VHA-1, 2, 3 and 4 (V0 c subunits), VHA-16 (V0 d subunit), VHA-17 (V0 e subunit), VHA-13 (V1 A subunit), VHA-12 (V1 B subunit), VHA-11 (V1 C subunit), VHA-14 (V1 D subunit), VHA-8 (V1 E subunit), VHA-9 (V1 F subunit), VHA-10 (V1 G subunit), and VHA-15 (V1 H subunit), and also two v-ATPase transporting accessory proteins, VHA-19 and VHA-20 (*Figure 2D*, *Supplementary file 1*). Among the V0 domain subunits, VHA-4, 5, 6, 7, and 16 and UNC-32 are enriched over 10-fold in all four replicates, VHA-1, 2, and 3 are enriched over 10-fold in three replicates and over 5-fold in one replicate, and the low abundant VHA-17 was only detected

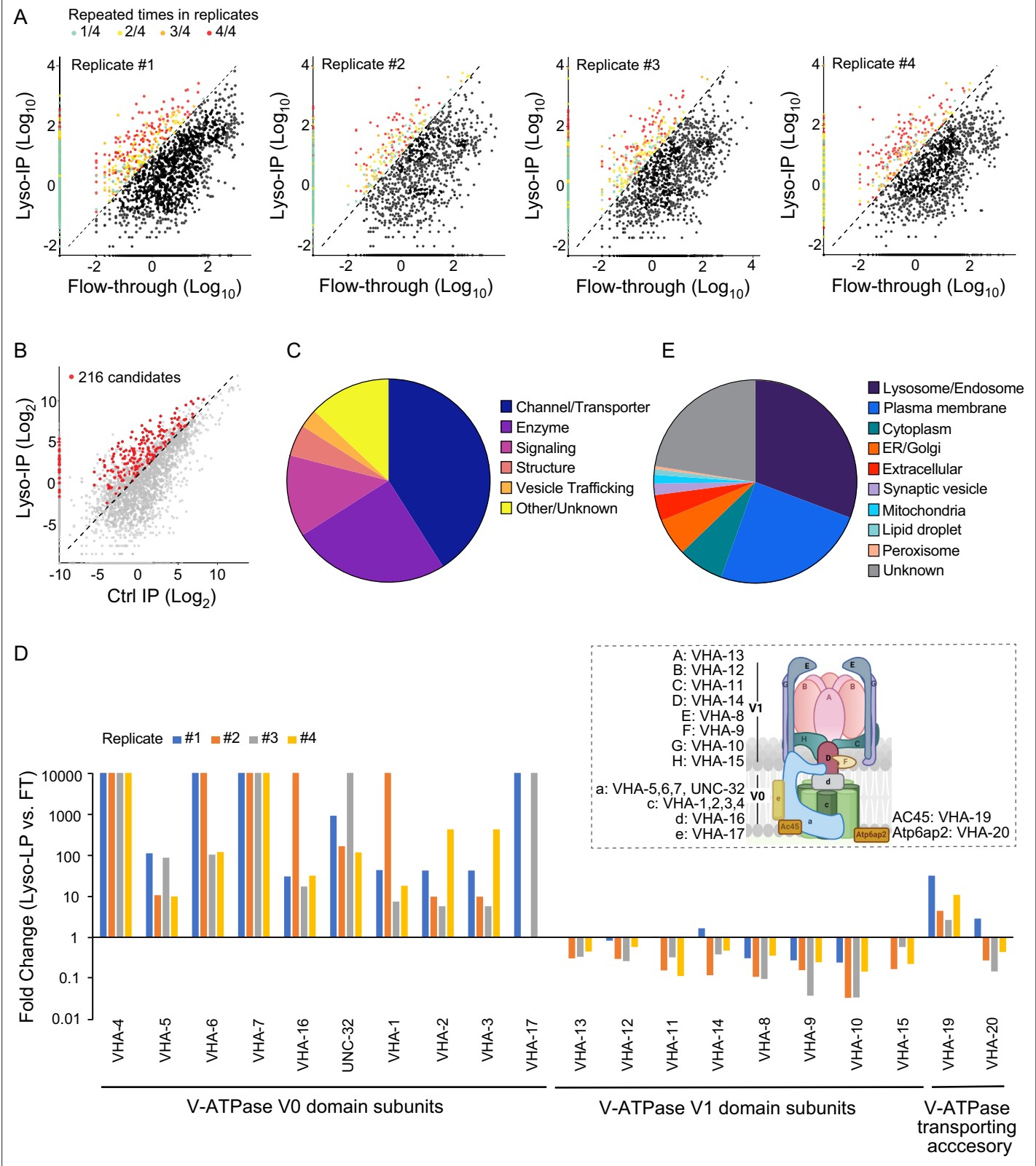

**Figure 2.** Systematic view of lysosome-enriched proteome. (**A**) Scatter plots showing candidate selection from four independent biological replicates in proteomics analyses. Proteins with at least 10-fold higher levels in Lyso-IP samples than in flow-through (FT) controls are highlighted with different colors based on repeated times in four replicates. (**B**) Scatter plot showing candidate selection with normalization to non-tagged controls using wild-type worms. 216 proteins with over twofold higher levels in Lyso-IP samples than in non-tagged controls are highlighted in red. (**C**) Pie chart showing

*Figure 2 continued on next page*

*Figure 2 continued*

molecular function categories of lysosome-enriched proteins. (**D**) The lysosomal enrichment ratio (Lyso-IP vs. FT) for each subunit of lysosomal vacuolar ATPase (v-ATPase) in four independent replicates is shown. Inserted scheme showing lysosomal V-ATPase assembly. (**E**) Pie chart showing subcellular location categories of lysosome-enriched proteins.

The online version of this article includes the following figure supplement(s) for figure 2:

**Figure supplement 1.** Pie chart showing the proportion of LMP-1 Lyso-IP candidates from wild-type (WT) worms with mammalian homologs.

in two replicates, with more than 10-fold enrichments in both (*Figure 2D*). The VHA-19 transporting accessory protein is enriched over 10-fold in two replicates and less than 5-fold in two replicates (*Figure 2D*). In contrast, for the subunits of the V1 domain and the VHA-20 transporting accessory protein, they show no enrichment in the purified lysosomes compared to the flow-through controls (*Figure 2D*). These results suggest that the free form of the V1 domain and the associated form bound with the V0 domain at lysosomes both exist under well-fed condition in WT worms.

In addition to 30.7% of proteins with known lysosome/endosome localization, the lysosome-enriched proteome includes a small portion of proteins localized to other cellular organelles, ER/Golgi (6.0%), mitochondria (1.4%), peroxisome (0.4%), lipid droplet (0.9%), and synaptic vesicle (1.8%) (*Figure 2E*). On the other hand, there is a large portion of proteins with annotated plasma membrane localization (24.8%) (*Figure 2E*). Many of these plasma membrane proteins are receptors that are known to be subject to endocytosis and subsequent recycling lysosomal degradation, such as INA-1/integrin alpha-6 (*De Franceschi et al., 2015*), VER-3/vascular endothelial growth factor receptor (*Ewan et al., 2006*), PTC-1/protein patched receptor (*Gallet and Therond, 2005*), and IGLR-2/leucine-rich repeat-containing G-protein-coupled receptor (*Snyder et al., 2013*; *Supplementary file 1*; *Braulke and Bonifacino, 2009*). We also identified proteins involved in the endocytosis process, including low-density lipoprotein receptor-related proteins, LRP-1 (*Grant and Hirsh, 1999*) and arrestin domain-containing proteins, ARRD-13 and ARRD-18 (*Kang et al., 2014*; *Supplementary file 1*) that mediate the internalization of plasma membrane receptors (*Ma et al., 2002*). Thus, the lysosome-enriched proteome also reveals membrane receptor proteins that undergo recycling through the endo-lysosomal system.

## Profile lysosome-enriched proteome heterogeneity among different tissues

Lysosomes are known as a heterogeneous population of vesicles, differing in their size, shape, pH, and cellular distribution. They broadly exist in all tissues of an organism and play diverse roles in a tissue-specific manner. To examine how lysosome-enriched proteomes exhibit heterogeneity among different tissues, we have generated four transgenic strains that overexpress LMP-1 Lyso-Tag specifically in neurons, muscle, intestine, and hypodermis using tissue-specific promoters, *unc-119*, *myo-3*, *ges-1*, and *col-12*, respectively (*Figure 3A*). Using these transgenic strains, we purified lysosomes in a tissue-specific manner and conducted proteomic profiling. The correlation analysis shows good reproducibility among three independent biological replicates (*Figure 3—figure supplement 1A-D*).

Unlike the whole-body Lyso-IP, the flow-through samples from tissue-specific Lyso-IP contain not only non-lysosomal fractions from the targeted tissue but also lysosomes from non-targeted tissues. Thus, these flow-through samples cannot be simply used as controls to determine the enrichment of proteins at the lysosome in the targeted tissue. To assess tissue-specific changes, we have normalized the level of each identified protein to the level of LMP-1 in the same replicate, and then compared the normalized ratio between the whole-body Lyso-IP and the tissue-specific Lyso-IP (*Figure 3B–E*, *Supplementary file 2*). We found that among the 216 proteins identified from the whole-body Lyso-IP, 85 of them show comparable ratios between the whole-body Lyso-IP and the four tissue-specific Lyso-IPs (*Figure 3F*, Group I), suggesting relative homogenous lysosomal enrichments of these proteins among different tissues. Nine of them were completely absent in the tissue-specific Lyso-IP, which may be related to their low abundance (*Figure 3F*, Group IV).

Furthermore, there are 122 proteins that exhibited significant differences in their enrichments between the whole-body Lyso-IP and the tissue-specific Lyso-IPs ($p < 0.05$), 56 of them (Group II) showing an increase in the tissue-specific Lyso-IPs while the other 66 (Group III) showing a decrease (*Figure 3F*). One of the candidates in Group II is Y58A7A.1, a copper uptake transporter, that shows a

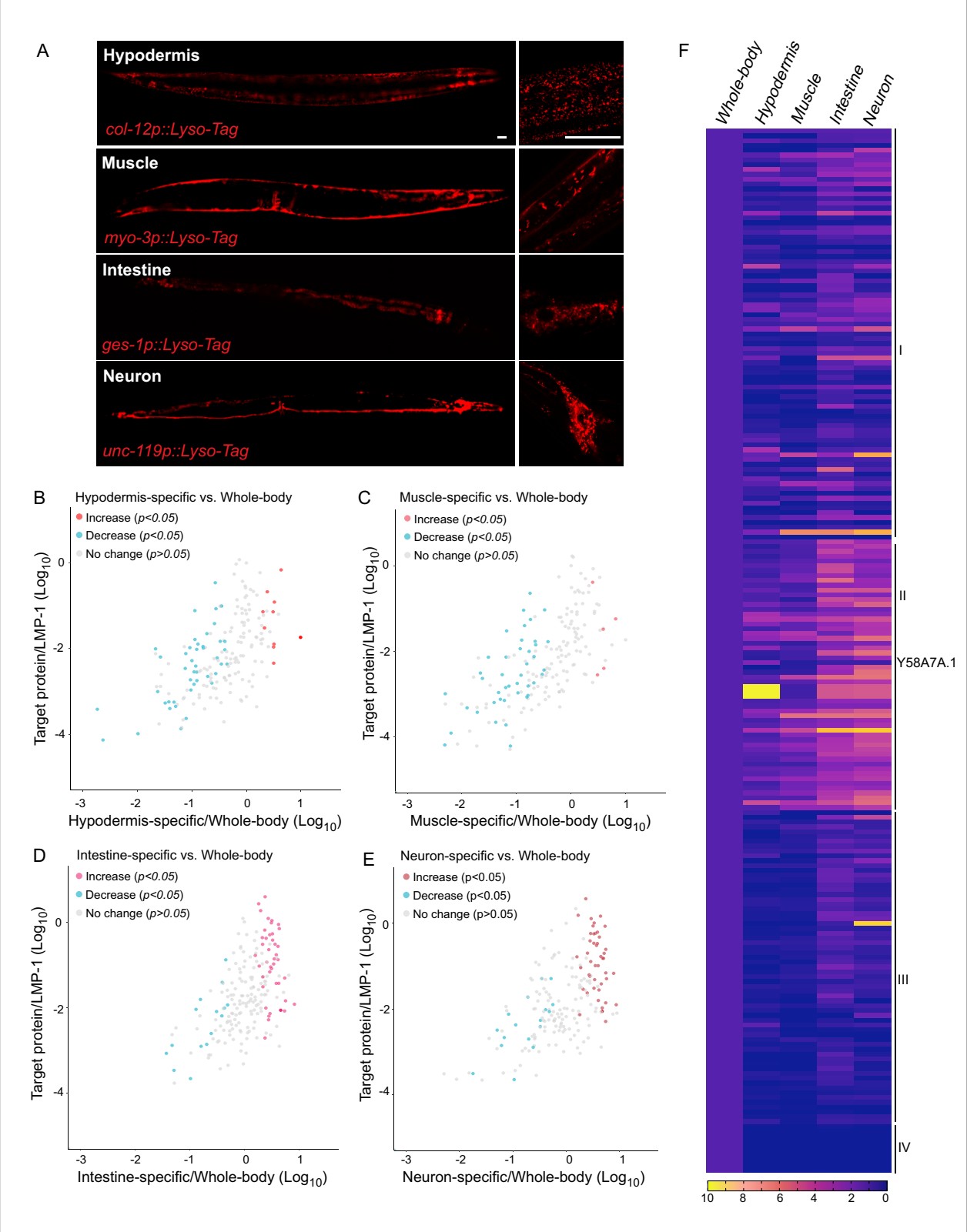

**Figure 3.** Lysosomal proteome heterogeneity across tissues. (**A**) Example images of transgenic strains carrying Lyso-Tag (LMP-1::RFP-3×HA) driven by four different tissue-specific promoters. Scale bar = 20 µm. Scatter plot showing the relative enrichment ratio for each of 216 lysosome-enriched proteins identified from whole-body LMP-1 Lyso-IP in comparison with tissue-specific LMP-1 Lyso-IPs, hypodermis (**B**), muscle (**C**), intestine (**D**), and neuron (**E**). *X*-axis, enrichment ratio tissue-specific vs. whole-body; *Y*-axis, normalized protein abundance over LMP-1; each dot represents the average of three

*Figure 3 continued on next page*

*Figure 3 continued*

replicates. (**F**) Heatmap showing the relative enrichment of 216 lysosome-enriched proteins identified from whole-body LMP-1 Lyso-IP in comparison with tissue-specific LMP-1 Lyso-IPs. Group I, comparable ratios between whole-body and tissue-specific Lyso-IPs; Group II, increase in tissue-specific Lyso-IPs (p < 0.05 by Student's *t*-test); Group III, decrease in tissue-specific Lyso-IPs (p < 0.05 by Student's *t*-test); Group IV, absent in tissue-specific IPs.

The online version of this article includes the following figure supplement(s) for figure 3:

**Figure supplement 1.** Tissue-specific Lyso-IPs and candidate imaging.

higher ratio in the hypodermis (**Figure 3F**). Copper transporters are crucial players in various biological processes and copper dysfunction contributes to oxidative stress, impaired respiration and enzymic activities, and disease progression. To validate whether Y58A7A.1 is a copper transporter specifically localized at the lysosome in the hypodermis, we generated a CRISPR knock-in line where the endogenous Y58A7A.1 is tagged with mNeonGreen. Using this line, we confirmed the hypodermis-specific lysosomal localization of Y58A7A.1 (**Figure 3—figure supplement 1E**).

These results show that the lysosomal proteome exhibits heterogeneity among different tissues within the organism, which may be related to the metabolic status in each tissue and, consequently, contribute to the specific activities and signaling effects of the tissue. Our studies provide a list of candidates for further investigation into the tissue-specific regulation of lysosomal metabolism and signaling.

## Lysosome-enriched proteome alterations associate with different pro-longevity mechanisms

Considering the emerging role of lysosomes as a cellular hub to integrate protein signals and regulate longevity, we next examined whether the protein composition of lysosomes exhibits heterogeneity in association with different longevity mechanisms. To this end, we crossed LMP-1 *LysoTg* with four different long-lived strains: the *lipl-4* transgenic strain (*lipl-4 Tg*) that constitutively expresses a lysosomal acid lipase (**Folick et al., 2015**), the loss-of-function mutant of *daf-2* (*daf-2(lf)*) that encodes the insulin/IGF-1 receptor (**Kenyon et al., 1993**; **Martins et al., 2016**), the loss-of-function mutant of *isp-1* (*isp-1(lf)*) that reduces mitochondrial electron transport chain complex III activity (**Feng et al., 2001**), and the *glp-1* loss-of-function mutant (*glp-1(lf)*) that has a defective germline at 25°C non-permissive temperature (**Berman and Kenyon, 2006**; **Figure 4A**). We then conducted Lyso-IP proteomic analyses and compared lysosome-enriched proteomes between WT and long-lived strains. The correlation analysis shows good reproducibility among three independent biological replicates (**Figure 4—figure supplement 1A-D**), and the PCA analysis shows a clear separation between Lyso-IP replicates and flow-through controls (**Figure 4—figure supplement 1E,F**).

In the *lipl-4 Tg* worms, we have identified 449 lysosome-enriched proteins (**Supplementary file 3**), and 176 of them overlap with the candidates from WT worms (**Figure 4B**). Thus, 82% of proteins enriched on WT lysosomes are also enriched on *lipl-4 Tg* lysosomes; however, 61% of proteins enriched on *lipl-4 Tg* lysosomes are absent in WT lysosomes (**Figure 4B**). In parallel, 259 lysosome-enriched proteins were identified in the *daf-2(lf)* mutant using LMP-1 Lyso-IP (**Supplementary file 4**), 147 of them overlapping with the LMP-1 Lyso-IP candidates from WT worms, 197 of them overlapping with the LMP-1 Lyso-IP candidates from the *lipl-4 Tg* worms, and 55 unique to the *daf-2(lf)* mutant (**Figure 4C**). In the *isp-1(lf)* mutant, we identified 177 lysosome-enriched proteins (**Supplementary file 5**). Among them, 26 candidates are unique to the *isp-1(lf)* mutant, while 107, 135, and 126 candidates overlap with those in the WT, *lipl-4 Tg*, and *daf-2(lf)* worms, respectively (**Figure 4C**). Meanwhile, 200 lysosome-enriched proteins were identified in the *glp-1(lf)* mutant (**Supplementary file 6**). When compared to the control worms growing at the same 25°C temperature (**Supplementary file 7**), 43 were unique to the *glp-1(lf)* mutant, while 157 overlapped with WT candidates (**Figure 4—figure supplement 1G**). Overall, there is only one lysosome-enriched candidate shared among all four long-lived strains but absent from the WT lysosome-enriched proteome (**Figure 4—figure supplement 1H**), suggesting that distinct pro-longevity mechanisms influence the protein composition of the lysosome in their specific ways. Furthermore, for the long-lived *daf-2(lf)*, *isp-1(lf)*, and *glp-1(lf)* worms, the overlaps of their lysosome-enriched proteome with the WT are 57%, 60%, and 78.5% (**Figure 4C**, **Figure 4—figure supplement 1G**), respectively. These percentages are higher than the 39% overlap observed between the long-lived *lipl-4 Tg* worms and the WT (**Figure 4B**). These results support that

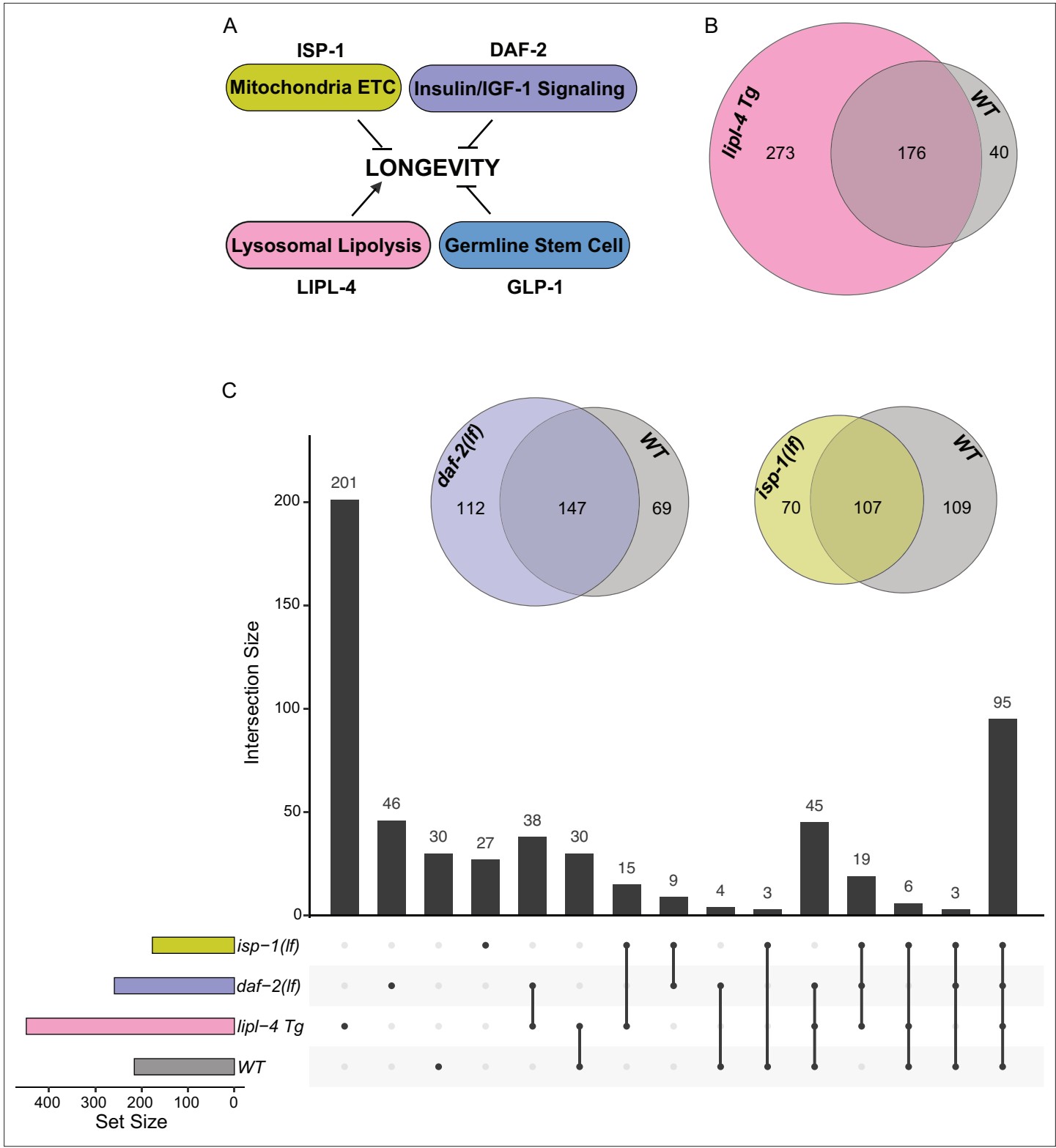

**Figure 4.** Lysosomal proteome in different pro-longevity models. (**A**) Scheme showing four different longevity regulatory mechanisms used in this study. Loss-of-function mutants (lf) of *isp-1*, *daf-2*, and *glp-1* reduce mitochondrial electron transport chain (ETC) complex III, insulin/IGF-1 signaling, and germline stem cell proliferation, respectively, leading to lifespan extension; while increasing lysosomal lipolysis by *lipl-4* transgenic overexpression (*lipl-4 Tg*) promotes longevity. (**B**) Venn diagram showing the overlap between the lysosome-enriched proteomes from wild-type (WT) and *lipl-4 Tg* worms.

*Figure 4 continued on next page*

Figure 4 continued

(C) Upset graph showing the distribution and overlap of lysosome-enriched proteins across the four pro-longevity models. Inserted Venn diagram showing the overlaps between the lysosome-enriched proteomes of WT worms and the long-lived *daf-2(lf)* and *isp-1(lf)* mutants.

The online version of this article includes the following figure supplement(s) for figure 4:

**Figure supplement 1.** Lyso-IP analyses from different long-lived strains.

increased lysosomal lipolysis leads to bigger changes on lysosomal protein composition than other pro-longevity mechanisms.

In the *lipl-4 Tg*, *daf-2(lf)*, and the *glp-1(lf)* lysosome-enriched proteomes, we found the enrichment of autophagosome proteins and proteins that mediate the fusion between autophagosomes and lysosomes, including ATG-9/ATG9A (*lipl-4 Tg*, *daf-2(lf)*, and *glp-1(lf)*), SQST-1/SQSTM1 (*daf-2(lf)* and *glp-1(lf)*), EPG-7/RB1CC1 (*lipl-4 Tg*), VAMP-7/VAMP8 (*lipl-4 Tg*), and Y75B8A.24/PI4KIIα (*lipl-4 Tg*) (*Figure 5A*), which is consistent with the previously reported induction of autophagy in these long-lived conditions (*Lapierre et al., 2011*; *Nakamura and Yoshimori, 2018*; *O'Rourke et al., 2013*).

In addition, we found that the Ragulator complex, LMTR-2/LAMTOR2, LMTR-3/LAMTOR3, and LMTR-5/LAMTOR5, that serves as a scaffold for the activation of mTORC1 and AMPK (*Zhang et al., 2014*), shows a higher enrichment at the lysosome from the *lipl-4 Tg* worms than WT (*Figure 5A*). However, such increased enrichments were not detected in the lysosome from the *daf-2(lf)*, *isp-1(lf)*, or *glp-1(lf)* mutant (*Figure 5A*). It is known that the Ragulator complex mediates the lysosomal activation of AMPK (*Zhang et al., 2014*). There are two homologs of AMPK catalytic units, AAK-1 and AAK-2 in *C. elegans*. We found that AAK-2 is enriched more than 10-fold in the Lyso-IP samples compared to the flow-through controls from the *lipl-4 Tg* worms, but it is only detected in the flow-through controls from WT worms (*Figure 5B*). Likely due to its low abundance, AAK-1 was detected twice in the Lyso-IP samples from the *lipl-4 Tg* worms but once only in the flow-through sample from WT worms (*Figure 5B*). On the other hand, AAK-1 and AAK-2 were not present in the lysosome-enriched proteome from the *daf-2(lf)*, *isp-1(lf)*, or *glp-1(lf)* mutant (*Figure 5B*). These results suggest that AMPK is specifically recruited to the lysosomal surface in the *lipl-4 Tg* worms, which may contribute to the pro-longevity effect. To test this idea, we inactivated AMPK using the *aak-2* loss-of-function mutant together with the *aak-1* RNA interference (RNAi) knockdown. We found that the AMPK inactivation reduces the lifespans of the *lipl-4 Tg* and WT worms by 29% and 17%, respectively, and suppresses the lifespan extension caused by *lipl-4 Tg* from 72% to 48% (*Figure 5C*, *Supplementary file 8*). Thus, *aak-1* and *aak-2* are partially responsible for the lifespan extension caused by *lipl-4 Tg*. For the AMPK catalytic subunits, It is known that the activation of AMPK displays high spatial specificity in the cell when responding to different upstream stimuli (*Khan and Frigo, 2017*). In *C. elegans*, it was previously shown that AAK-2 mediates the longevity effect conferred by the *daf-2(lf)* mutant (*Apfeld et al., 2004*). Our results indicate that this regulation might not be associated with the lysosomal activation of AMPK, and the spatial specificity of AMPK activation at different subcellular compartments may be linked with different longevity mechanisms.

Moreover, compared to WT lysosomes, the enrichment of lysosomal v-ATPase is higher in *lipl-4 Tg* lysosomes, especially for the V0 subunits, VHA-1, 2, 3 the V1 subunits, VHA-11 and VHA-15, and the v-ATPase transporting accessory proteins, VHA-19 and VHA-20 (*Figure 5A*). There are also 13 lysosomal channels/transporters, including T14B4.3/ATP6AP2, the proton-translocating ATPases required for the v-ATPase-mediated lysosomal acidification (*Cruciat et al., 2010*) and CLH-6/CLCN7, the H(+)/Cl(−) exchange transporter mediating the acidification of the lysosome (*Graves et al., 2008*; *Nicoli et al., 2019*), and 16 lysosomal hydrolases that are specifically associated with the *lipl-4 Tg* lysosomes (*Figure 5A*). However, none of these components exhibit increased enrichments in the lysosome from the *daf-2(lf)*, *isp-1(lf)*, or *glp-1(lf)* mutant (*Figure 5A*). Together, these results suggest that the proportion of mature acidic lysosomes may be increased in the *lipl-4 Tg* worms, which may lead to increased autophagy, the lysosomal activation of AMPK, and consequently the induction of longevity.

## Enhanced lysosome–nucleus proximity mediates longevity responding to lysosomal lipolysis

It is known that the luminal pH of lysosomes is affected by their cellular position, with perinuclear lysosomes being more acidic (*Johnson et al., 2016*). In the cell, mobile lysosomes can change their

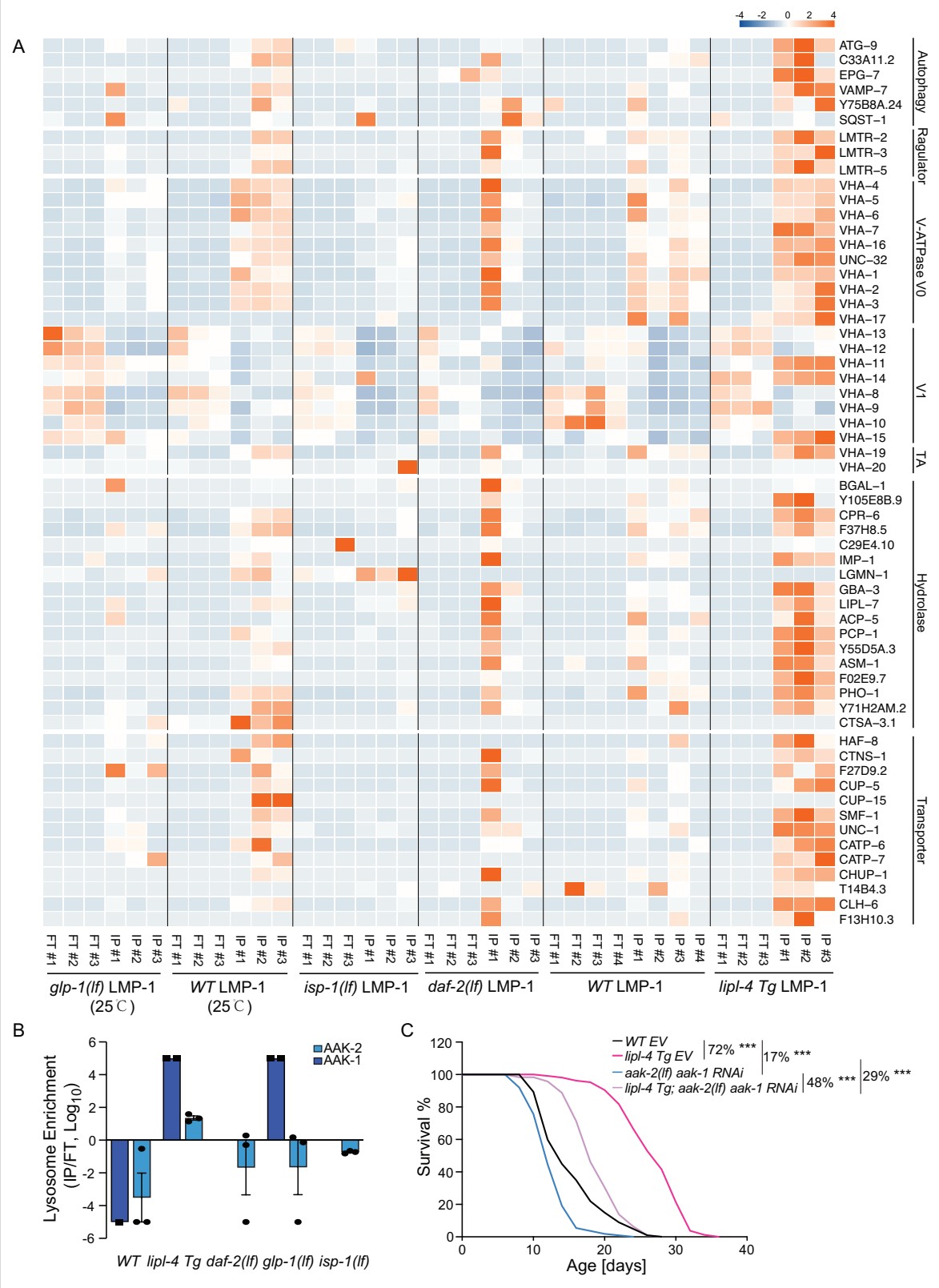

**Figure 5.** Increased enrichment of lysosomal proteins upon lysosomal lipolysis. (**A**) Normalized protein levels (*z*-score across samples) of autophagy-related components, mTORC1 signaling factors, lysosomal v-ATPase V0, V1, and transporting accessory (TA) subunits, lysosomal hydrolases and transporter proteins from LMP-1 Lyso-IP proteomic analyses of wild-type (WT), *lipl-4 Tg*, *daf-2(lf)*, and *isp-1(lf)* worms grown at 20°C and WT and *glp-1(lf)* worms grown at 25°C. (**B**) The lysosomal enrichment ratio (Lyso-IP vs. FT) for two homologs of AMP-activated protein kinase (AMPK) catalytic subunits,

*Figure 5 continued on next page*

*Figure 5 continued*

AAK-1 and AAK-2 in WT, *lipl-4 Tg*, *daf-2(lf)*, *isp-1(lf)*, and *glp-1(lf)* worms. (**C**) Reduction of AMPK using the loss-of-function mutant of *aak-2, aak-2(lf)* together with *aak-1* RNAi knockdown decreases lifespan by 17% and 29% in the WT and *lipl-4 Tg* background, respectively. As a result, the lifespan extension caused by *lipl-4 Tg* is reduced from 72% to 48%. ***p < 0.001 by Log-rank test. The lifespan data are also in **Supplementary file 8**.

distribution along the perinuclear–peripheral axis in response to different nutrient signals and metabolic status (**Ballabio and Bonifacino, 2020**; **Pu et al., 2016**). Interestingly, when analyzing the LMP-1 lysosome-enriched proteome in the *lipl-4 Tg* worms, we found an enrichment of nucleus-localized proteins (**Figure 6A**), including two nucleoporin proteins NPP-6/Nup160 and NPP-15/Nup133 in the Nup160 complex that localizes at the basket side of the nuclear pore (**Figure 6B**; **Vasu et al., 2001**). Such enrichment of nucleoporin proteins was not found in the LMP-1 lysosome-enriched proteome of the *daf-2(lf)*, *isp-1(lf)*, or *glp-1(lf)* long-lived mutant. We thus hypothesize that LIPL-4-induced lysosomal lipolysis may increase the proximity between lysosomes and the nucleus, accompanied by an increase in lysosomal acidity.

To test this hypothesis, we imaged lysosomal positions in intestinal cells where *lipl-4* is expressed. Using a dual reporter strain expressing both lysosomal LMP-1::RFP fusion and nucleus-localized GFP, we found that lysosomes exhibit a dispersed pattern in the intestinal cell of WT worms (**Figure 6C**). However, in the *lipl-4 Tg* worms, lysosomes are clustered in the perinuclear region (**Figure 6C**), supporting the hypothesis that the proximity between lysosomes and the nucleus is increased. To quantitatively measure this change in lysosomal positioning, we analyzed the RFP fluorescent signal distribution in intestinal cells (**Figure 6—figure supplement 1A**). We found the perinuclear and peripheral distribution of lysosomes in the *lipl-4 Tg* worms is significantly increased and decreased, respectively, compared to WT worms (p < 0.01, **Figure 6D**, **Figure 6—figure supplement 1B**). In contrast, such perinuclear clustering is not observed in intestinal cells of the *daf-2(lf)* mutant (**Figure 6E, F**, **Figure 6—figure supplement 1C**).

Moreover, we found that the RNAi knockdown of *npp-6* suppresses the lifespan extension in the *lipl-4 Tg* worms (**Figure 6G, H**) but does not affect the lifespan extension in the *daf-2(lf)* (**Figure 6I, J**) or the *isp-1(lf)* mutant (**Figure 6—figure supplement 1D, E**). These results suggest that the nucleoporin protein NPP-6 is specifically involved in the regulation of lysosomal LIPL-4-induced longevity. Given the importance of nucleoporin in nuclear transport, we further test whether nuclear import and/or export may play a role in regulating *lipl-4 Tg*-induced longevity. To this end, we knocked down *xpo-1* and *ima-3*, which encodes Exportin-1 and Importin-α, mediating nuclear export and import, respectively, by RNAi. We found that the RNAi inactivation of *ima-3*, but not *xpo-1* suppresses the lifespan extension caused by *lipl-4 Tg* (**Figure 6—figure supplement 1F-H** and **Supplementary file 8**). These results suggest that the increased proximity between lysosomes and the nucleus may facilitate the nuclear import of *lipl-4 Tg*-induced lysosomal retrograde signals to promote longevity.

## Cystinosin-positive mature lysosomes enrich specific lysosomal proteins

The analysis of the candidates specifically detected in the *lipl-4 Tg* worms suggests that the proportion of mature lysosomes may affect lysosomal protein composition. Although LMP-1 is a well-established lysosomal protein marker and highly abundant on the lysosomal surface, it can be also detected in late endosomes and sometimes in early endocytic compartments. With the hope to profile proteins enriched in mature lysosomes, we chose CTNS-1, the *C. elegans* lysosomal cystine transporter Cystinosin that is a well-established marker of mature lysosomes (**Gahl et al., 1982**; **Jonas et al., 1982**; **Kalatzis et al., 2001**). Using CRISPR knock-in lines with endogenous CTNS-1 and LMP-1 tagged with wrmScarlet and mNeonGreen, respectively, we found that CTNS-1 and LMP-1 signals show only partial overlap in the intestine, muscle, hypodermis, and neurons (**Figure 7—figure supplement 1**). We then generated a transgenic strain expressing CTNS-1 tagged with both 3×HA and RFP (CTNS-1 *lysoTg*). Fluorescence imaging of RFP confirmed the lysosomal localization of the CTNS-1 fusion protein in live organisms (**Figure 7A**). Using this transgenic strain, we followed the same Lyso-IP and MS profiling pipeline. The correlation analysis shows good reproducibility among three independent biological replicates (**Figure 7—figure supplement 2A**), and the PCA analysis indicates a clear separation between Lyso-IP samples and flow-through controls (**Figure 7—figure supplement 2B**). Using the same selection criteria, we identified 293 candidates whose levels are enriched at least 10-fold in the purified lysosomes than those in the flow-through controls among all independent biological

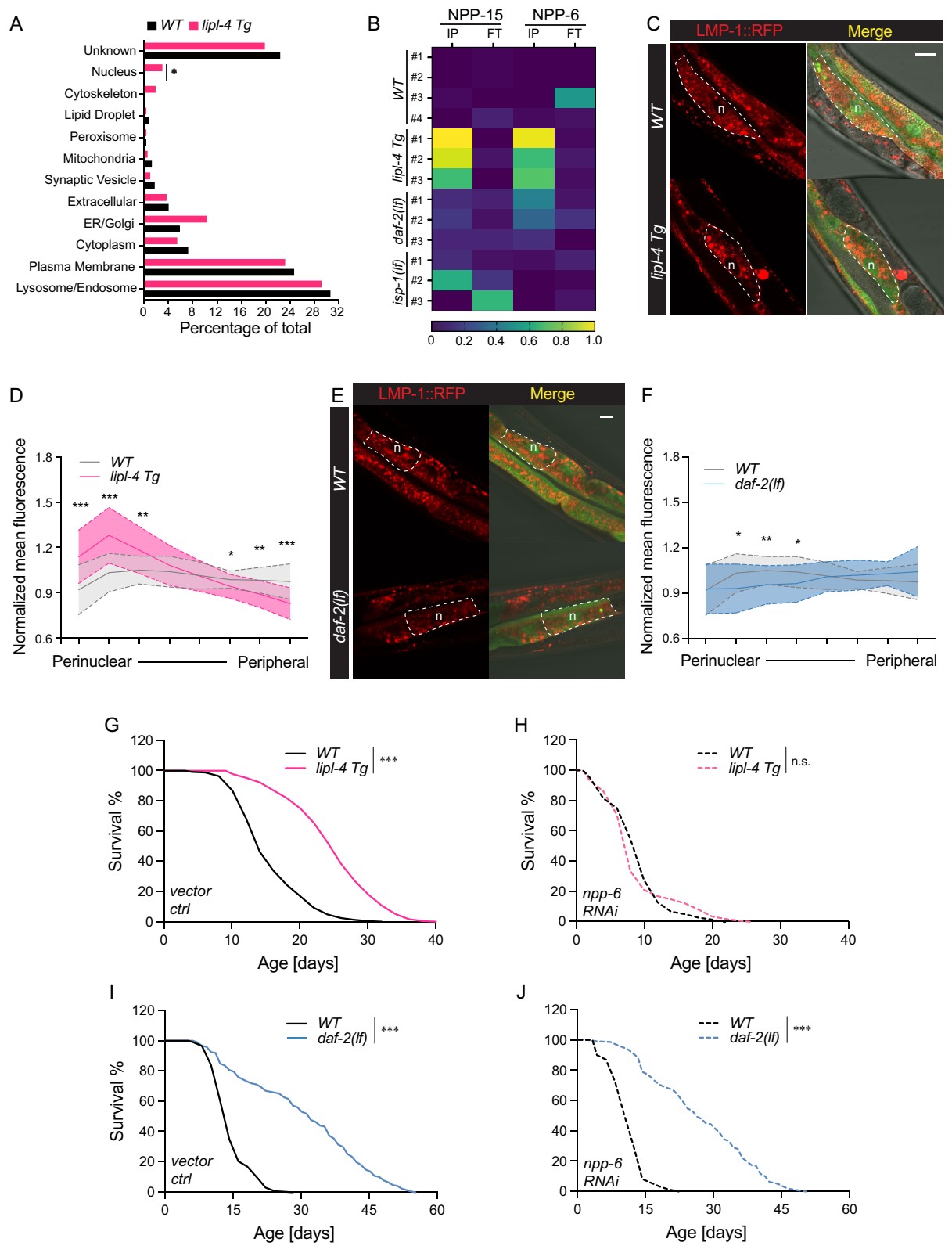

**Figure 6.** Enhanced lysosome–nucleus proximity contributing to longevity. (**A**) The percentage of proteins with different subcellular localization is compared between lysosome-enriched proteomes from wild-type (WT) and *lipl-4 Tg* worms. *p = 0.019 by two-sample test for equality of proportions. (**B**) Heatmap showing the average levels of nucleoporin proteins NPP-6 and NPP-15 in Lyso-IP (IP) and flow-through (FT) samples from WT, *lipl-4 Tg*, *daf-2(lf)*, and *isp-1(lf)* worms. Representative images of intestinal cells in WT, *lipl-4 Tg* (**C**), and *daf-2(lf)* (**E**) worms carrying LMP-1::RFP-3×HA and

*Figure 6 continued on next page*

*Figure 6 continued*

nucleus-enriched GFP, showing the accumulation of lysosomes around the perinuclear region in the *lipl-4 Tg* but not *daf-2(lf)* worms. Dashed lines circle intestinal cells and n marks the nucleus. Scale bar = 20 µm. Line graph showing the spatial distribution of lysosomes from the nuclear to peripheral region quantified by normalized regional RFP fluorescence signals in intestinal cells of WT, *lipl-4 Tg* (**D**), and *daf-2(lf)* (**F**) worms. N = 50 WT/33 *lipl-4 Tg*, 33 WT/28 *daf-2(lf)*. Data are represented as mean ± standard deviation (SD). p values for (**D**) (from left to right): $1.23 \times 10^{-7}$, $2.25 \times 10^{-5}$, 0.00322, 0.368, 0.273, 0.0447, 0.00268, $1.20 \times 10^{-5}$; p values for (**F**) (from left to right): 0.633, 0.0211, 0.00259, 0.0359, 0.767, 0.151, 0.106, 0.0671. *lipl-4 Tg* worms show lifespan extension compared to WT worms (**G**), which is fully suppressed by RNAi knockdown of *npp-6* (**H**). ***p < 0.001, n.s. p > 0.05 by Log-rank test. *daf-2(lf)* worms show lifespan extension compared to WT worms (**I**), which is not affected by *npp-6* RNAi knockdown (**J**). ***p < 0.001 by Log-rank test. The lifespan data are also in **Supplementary file 8**.

The online version of this article includes the following figure supplement(s) for figure 6:

**Figure supplement 1.** Lysosomal positioning in longevity regulation.

replicates and show over 2-fold enrichment compared to the non-tag controls (**Supplementary file 9**). There are 95 lysosome-enriched proteins shared between the LMP-1 and the CTNS-1 Lyso-IP proteomic profiling datasets (**Figure 7B**, **Supplementary file 9**), and 47 of these shared proteins are annotated with lysosomal localization (**Supplementary file 9**). We have also crossed the CTNS-1 *lysoTg* strain with the *lipl-4 Tg*, *daf-2(lf)*, and *glp-1(lf)* worms and then conducted Lyso-IP proteomic profiling. However, the pull-down efficiency was very low in these long-lived worms, which prevented us from identifying proteins unique to CTNS-1 Lyso-IP in those conditions.

In WT worms, the proportions of the identified proteins with different categories of subcellular annotation are comparable between LMP-1 and CTNS-1 Lyso-IP conditions (**Figures 7C, 2E**), and for proteins with lysosomal annotation, the proportion is 25% and 28% in LMP-1 and CTNS-1 Lyso-IP, respectively (**Figure 7D**). However, among the 121 proteins only identified in LMP-1 Lyso-IP, there are only 8 with lysosomal annotation (7%); while for the 198 proteins only identified in CTNS-1 Lyso-IP, 35 are with lysosomal annotation and the proportion remains as 18% (**Figure 7D**).

Among the lysosomal proteins that are unique to CTNS-1 Lyso-IP, there are autophagosome proteins and proteins that mediate the fusion between autophagosomes and lysosomes, including ATG-9/ATG9A (**Popovic and Dikic, 2014**), C33A11.2/DRAM2 (**Crighton et al., 2006**), EPG-7/RB1CC1 (**Nishimura et al., 2013**), and VAMP-7/VAMP8 (**Diao et al., 2015**; **Itakura et al., 2012**; **Figure 7E**). Furthermore, the Ragulator complex components LMTR-2/3/5, the lysosomal amino acid transporter F13H10.3/SLC38A9 and the lysosomal calcium channel CUP-5/TRPML1 that regulate mTORC1 signaling (**Li et al., 2016**; **Rebsamen et al., 2015**; **Wang et al., 2015**; **Wyant et al., 2017**) exhibited a higher enrichment in the lysosome purified from WT worms using CTNS-1 Lyso-IP than using LMP-1 Lyso-IP (**Figure 7E**). To further confirm the increased enrichment of mTORC1 signaling components with CTNS-1 lysosomes, we generated a CRISPR knock-in line with endogenous LMTR-3 tagged with wrmScarlet to visualize its subcellular localization. After crossing this line with LMP-1::mNeonGreen and CTNS-1:: mNeonGreen knock-in lines, we found that LMTR-3 shows a complete overlap with CTNS-1 in the intestine, muscle and hypodermis (**Figure 7F**, **Figure 7—figure supplement 2C**), but it only shows a partial overlap with LMP-1 in the intestine, muscle and hypodermis (**Figure 7F**, **Figure 7—figure supplement 2D**).

Furthermore, when systemically examining 85 lysosome-related proteins that were previously annotated in *C. elegans* based on sequence homology (**Sun et al., 2020**), we found that 63 were detected in the proteomic profiling, while 22 were not detected likely due to their low abundance (**Figure 7G**). Many lysosomal hydrolases exhibit increased enrichments with CTNS-1 Lyso-IP. Interestingly, similar increased enrichments of these candidates were also observed in the LMP-1 Lyso-IP result using the *lipl-4 Tg* worms (**Figure 7E, G**). These results further support that the long-lived *lipl-4 Tg* worms carry more acidic lysosomes. The enrichment of cysteine proteases including CPR-6, LGMN-1, CPL-1, CPZ-1, and TAG-196 is consistent with that CTNS-1 is located at mature lysosomes as a cysteine transporter (**Gahl et al., 1982**; **Jonas et al., 1982**; **Kalatzis et al., 2001**). Together, we found that lysosome-enriched proteomes identified from both LMP-1 and CTNS-1 Lyso-IP consist of well-characterized lysosomal enzymes and integral membrane proteins as well as proteins that contribute to lysosomal signaling, dynamics and contact with other cellular compartments. Besides many known lysosomal proteins, various proteins that are not previously linked with lysosomes are now identified through these systematic analyses.

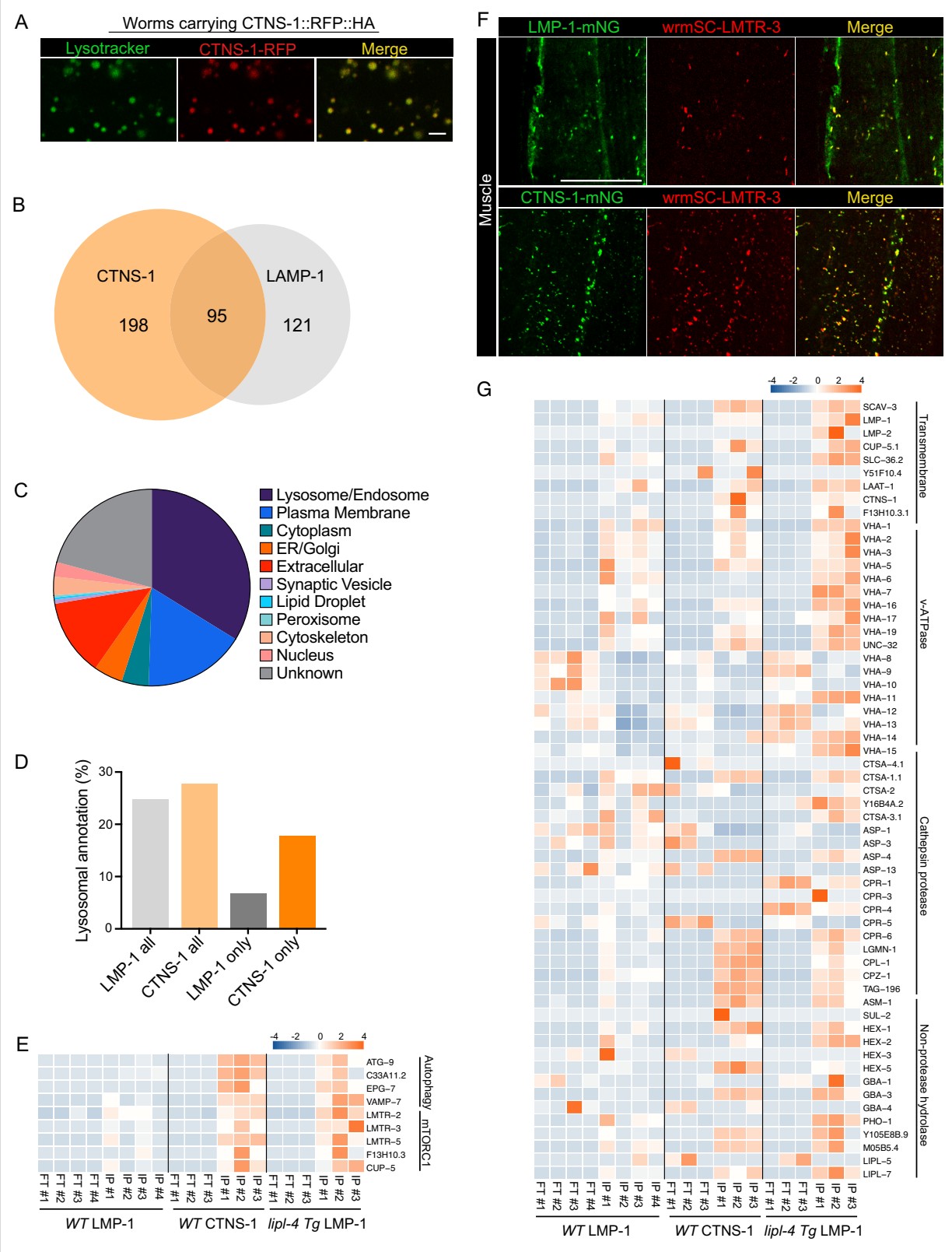

**Figure 7.** Lysosome-enriched proteome identified with Cystinosin. (**A**) Example images of transgenic strains carrying CTNS-1 Lyso-Tag (CTNS-1::RFP-3×HA) with LysoTracker staining to mark lysosomes in vivo. Scale bar = 5 μm. (**B**) Venn diagram showing the overlap between lysosome-enriched proteomes using LMP-1 Lyso-IP and CTNS-1 Lyso-IP. (**C**) Pie chart showing subcellular location categories of lysosome-enriched proteins. (**D**) The proportion of candidates with lysosomal localization annotation in different candidate groups. 'LMP-1 all' and 'CTNS-1 all', all candidates from LMP-1

*Figure 7 continued on next page*

*Figure 7 continued*

Lyso-IP and CTNS-1 Lyso-IP, respectively; 'LMP-1 only' and 'CTNS-1 only', candidates only identified from LMP-1 Lyso-IP or CTNS-1 Lyso-IP, respectively. (**E**) Normalized protein levels (z-score across samples) of autophagy-related components and mTORC1 signaling factors from CTNS-1 Lyso-IP proteomic analyses of wild-type (WT) worms and LMP-1 Lyso-IP proteomic analyses of WT and *lipl-4 Tg* worms. (**F**) Representative muscle images in the wrmScarlet::LMTR-3 knock-in line crossed with either LMP-1::mNeonGreen knock-in line or CTNS-1::mNeonGreen knock-in line. Scale bar = 20 μm. (**G**) Normalized protein levels (z-score across samples) of previously annotated lysosomal proteins from LMP-1 Lyso-IP proteomic analyses of WT and *lipl-4 Tg* worms and CTNS-1 Lyso-IP proteomic analyses of WT worms.

The online version of this article includes the following figure supplement(s) for figure 7:

**Figure supplement 1.** The colocalization between LMP-1::mNeonGreen and CTNS-1::wrmScarlet in different tissues.

**Figure supplement 2.** CTNS-1 Lyso-IPs and LMTR-3 imaging analyses.

## Lysosome-enriched proteins regulate different lysosomal activities

To understand the role of these newly identified lysosome-enriched proteins in regulating lysosomal functions, we have examined their effects on lysosomes using an RNAi screen based on LysoSensor fluorescence intensity. We focused on 95 lysosome-enriched proteins shared between LMP-1 and CTNS-1 Lyso-IPs and knocked down their coding genes by RNAi, and then used LysoSensor probes to stain lysosomes. From screening these 95 candidates (*Supplementary file 10*), we have identified five genes whose inactivation cause changes in LysoSensor signal intensity, and four of them have human homologs, including two lysosomal v-ATPase subunits, UNC-32/ATP6V0A and VHA-5/ATP6V0A, the lysosomal amino acid transporter SLC-36.2/SLC36A1 (SLC36A4), and a transmembrane protein R144.6/TMEM144 (*Supplementary file 10*, *Figure 8A–E*, *Figure 8—figure supplement 1*). We further examined their effects on the lysosomal number, size, and pH. We found that the RNAi knockdown of the two lysosomal v-ATPase subunits, UNC-32 and VHA-5, lead to decreased lysosomal numbers (*Figure 8F*), but an increase in the lysosomal size (*Figure 8G*). We also used fluorescence lifetime microscopy to measure the fluorescence lifetime of LysoSensor, which is negatively correlated with pH (*Deng et al., 2023*; *Lin et al., 2001*). Unexpectedly, we found that the RNAi knockdown of *unc-32* increases the fluorescence lifetime of LysoSensor, indicating a decrease in lysosomal pH (*Figure 8H*). We think this decrease is an attempt to compensate for the 2.5-fold reduction in the total number of lysosomes. Overall, *unc-32* inactivation compromises lysosomal v-ATPase and leads to a defect in lysosomal maturation. On the other hand, the RNAi knockdown of R144.6 did not affect lysosomal number or size but increased lysosomal pH (*Figure 8F–H*).

Unlike well-known lysosomal proteins, UNC-32, VHA-5, and SLC-36.2, the subcellular localization of the R144.6 protein remains unknown. R144.6 is a predicted carbohydrate transporter, and structural simulation using AlphaFold2 suggested it as a solute carrier family (SLC) transporter (*Figure 8I*). We generated a CRISPR knock-in line in which the endogenous R144.6 protein is fused with mNeonGreen and then stained these worms with LysoTracker Red to mark lysosomes. In the hypodermis, we found that mNeonGreen and LysoTracker Red signals overlap, confirming the lysosomal localization of this newly identified transmembrane protein from Lyso-IP (*Figure 8J*). On the other hand, its expression was not detected in the muscle, and in the intestine, its mNeonGreen signals did not overlap with LysoTracker Red, which is consistent with tissue-specific Lyso-IP analyses. R144.6 was enriched in the hypodermis-specific Lyso-IP at a similar level as in the whole-body Lyso-IP; however, it was not detected in the muscle- or intestine-specific Lyso-IP. These results further support the enrichment specificity of proteins at the lysosome in different tissues as well as the power of the Lyso-IP proteomic profiling in discovering new lysosomal proteins with functional significance.

## Discussion

Our studies reveal the heterogeneity of lysosomal protein composition that is associated with lysosomal status, tissue specificity, and organism longevity. Through systematic profiling of lysosome-enriched proteins under different conditions, we confirmed the induction of lysosome-related autophagy by different longevity-promoting pathways, unveiled increased cellular interaction between lysosomes and the nucleus upon the induction of lysosomal lipolysis and its contribution to longevity regulation, and underlined the importance of the spatial control of AMPK activation in regulating longevity. Our work provides not only methods for future studies to profile the dynamics of the lysosomal proteome

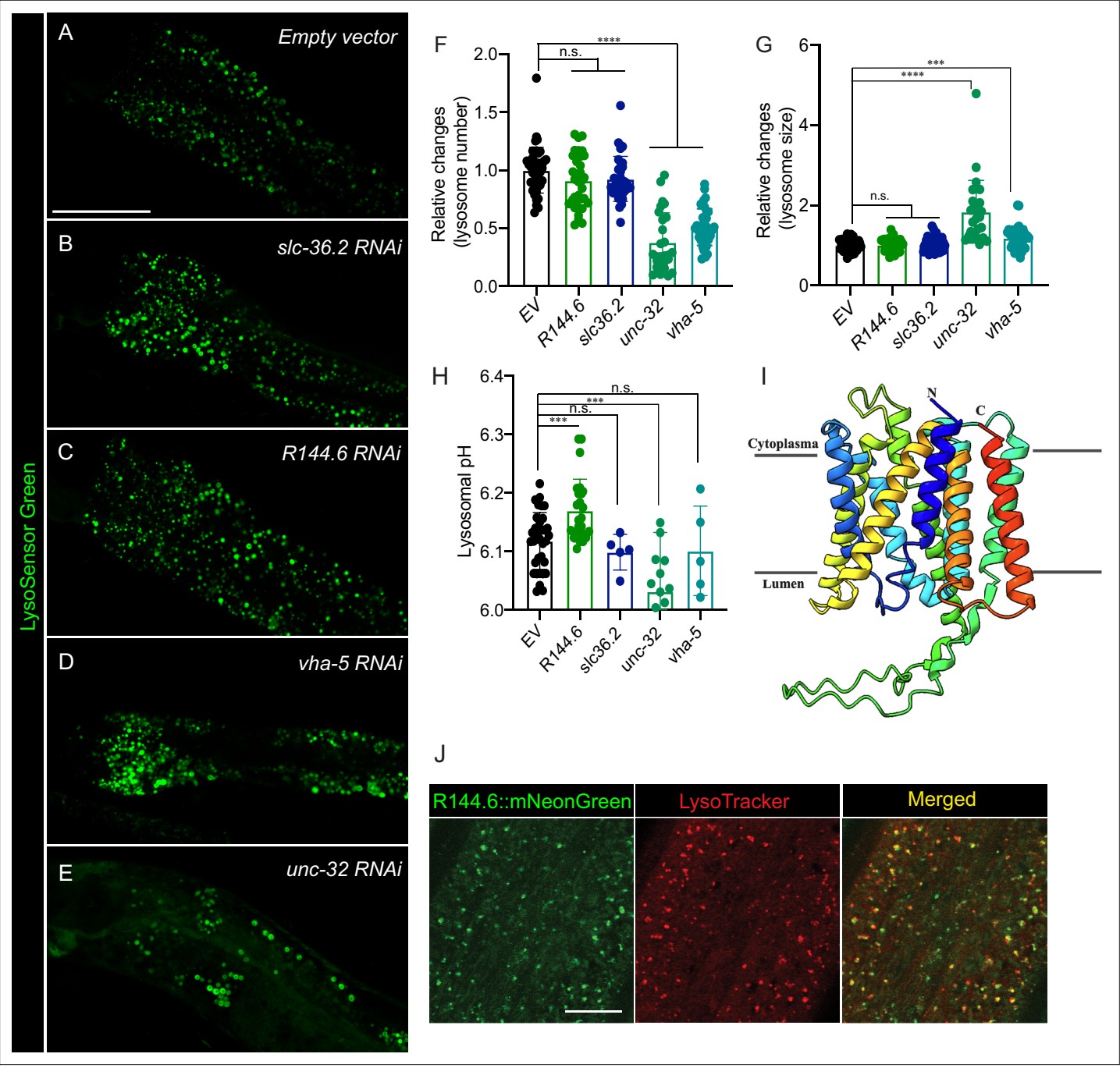

**Figure 8.** Lysosome-enriched proteins regulating lysosomal functions. Confocal fluorescence microscopy images of intestinal cells in worms stained with LysoSensor DND-189 and treated with *empty vector* (**A**), *slc36.2* RNAi (**B**), *R144.6* RNAi (**C**), *vha-5* RNAi (**D**), and *unc-32* RNAi (**E**). Scale bar = 50 µm. RNAi knockdown of *unc-32* or *vha-5* decreases the lysosome number (****p < 0.0001) (**F**) but increases the lysosome size (****p < 0.0001, ***p < 0.001) (**G**). The average lysosome number and size per pair of intestinal cells were quantified. Data are shown as mean ± standard deviation (SD). Student *t*-test (unpaired, two-tailed) was performed between the *empty vector* and RNAi-treated groups. At least three independent experiments with ~10 worms in each were performed for each condition. n.s. p > 0.05, (**H**) RNAi knockdown of *R144.6* and *unc-32* (***p < 0.001) increase and decrease lysosomal pH, respectively. Lysosomal pH was calculated based on LysoSensor's lifetime measured by Fluorescence Lifetime Microscopy. Data are shown as mean ± SD. Student's *t*-test (unpaired, two-tailed) was performed between the *empty vector* and RNAi-treated groups. Two independent experiments with at least five worms in each were performed in *R144.6* RNAi and *unc-32* RNAi conditions. The *vha-5* and *slc36.2* RNAi knockdown did not show significant changes in one replicate and were not retested with another replicate. n.s. p > 0.05. (**I**) The structure of the R144.6 protein predicted by AlphaFold2 supports it as a solute carrier family transporter. (**J**) Confocal fluorescence microscopy images show that mNeonGreen signals from endogenously tagged R144.6 colocalize with LysoTracker Red signals in the hypodermis. Scale bar = 10 µm.

*Figure 8 continued on next page*

*Figure 8 continued*

The online version of this article includes the following figure supplement(s) for figure 8:

**Figure supplement 1.** LysoSensor intensity quantification in five candidates.

in response to diverse physiological inputs, but also resources for understanding the vital contribution of these dynamics in modulating signal transduction, organelle crosstalk, and organism longevity.

These proteomic studies can provide hits for changes in the interaction between lysosomes and other organelles under different conditions. One example is the lysosome–nucleus interaction. In the Lyso-IP fraction from WT worms, we did not detect any proteins with sole nuclear localization; however, in the Lyso-IP fraction from the *lipl-4 Tg* or *daf-2(lf)* mutant worms, nuclear proteins were identified and the percentage of the increase over WT is significantly higher in the *lipl-4 Tg* worms (p < 0.05, *Figure 6A*, *Figure 6—figure supplement 1I*). Based on this finding, we discovered the previously unknown perinuclear accumulation of lysosomes in the *lipl-4 Tg* worms (*Figure 6C, D*) and further confirmed its importance for longevity regulation (*Figure 6G, H*). It has been shown previously that perinuclear lysosomes are more acidic than peripheral lysosomes (*Johnson et al., 2016*; *Webb et al., 2021*). Thus, the increase in perinuclear lysosomes may be associated with the increased proportion of mature lysosomes in the *lipl-4 Tg* worms, which is supported by the increased enrichments of lysosomal v-ATPase, channels/transporters, and hydrolases (*Figure 5A*). This increased distribution of lysosomes toward the perinuclear region could facilitate proteins and metabolites transporting from the lysosome to the nucleus through the nuclear pore and in turn their signaling effects. However, whether this perinuclear distribution of lysosomes is associated with an increase in the direct contact between lysosomes and the nucleus remains to be determined using technologies with higher spatial resolution such as electron microscopy imaging. On the other hand, we did not detect perinuclear accumulation of lysosomes in the *daf-2(lf)* mutant worms by cellular imaging, and the nuclear proteins detected through LMP-1 Lyso-IP from the *daf-2(lf)* mutant worms are mainly involved in RNA splicing. In yeast cells, defects in pre-mRNA processing have been associated with nucleophagy (*Léger-Silvestre et al., 2005*) that involves SQSTM1 and lysosomes (*Ivanov et al., 2013*; *Mijaljica and Devenish, 2013*). We thus speculate that the increased enrichment of nuclear proteins in the *daf-2(lf)* mutant worms may be associated with the induction of nucleophagy but not changes in lysosomal positioning.

mTORC1 and AMPK are key metabolic checkpoints that regulate anabolic and catabolic processes in mutually opposing ways. In sensing the lack of nutrients, AMPK signals activate the catabolic process while inhibiting the anabolic one. On the other hand, responding to nutrient availability, mTORC1 activation upregulates anabolic metabolism and promotes cell growth. Intriguingly, it is now known that both mTORC1 and AMPK are recruited to the lysosomal surface for activation, which requires the scaffold Ragulator complex that consists of LAMTOR subunits. We found that the Ragulator complex (LMTR-2, 3, and 5) shows a higher enrichment in the lysosome purified from WT worms using CTNS-1 Lyso-IP than using LMP-1 Lyso-IP (*Figure 7E*), and cellular imaging of endogenous LMTR-3 confirmed its much higher overlap with CTNS-1-positive lysosomes than with LMP-1-positive lysosomes (*Figure 7F*, *Figure 7—figure supplement 2C, D*). These results suggest a predominant association of the Ragulator complex with mature lysosomes, which could in turn determine the preference of mTORC1 and AMPK activation at the lysosomal surface. Alternatively, it would be also possible that the Ragulator complex carries a preference toward CTNS-1/Cystinosin-containing lysosomes, which would infer the interaction between lysosomal cysteine metabolism and mTORC1 signaling. Interestingly, previous studies show that Cystinosin co-immunoprecipitates with the Ragulator complex in mammalian cells (*Andrzejewska et al., 2016*), and in *Drosophila*, cysteine efflux from the lysosome via Cystinosin antagonizes mTORC1 signaling and upregulates the tricarboxylic acid cycle (*Jouandin et al., 2022*). Whether this inhibitory effect of lysosomal cysteine on mTORC1 is related to the preferential interaction between the Ragulator complex and Cystinosin would be an interesting question for future studies.

Both mTORC1 and AMPK have been implicated in the regulation of longevity across different species, being intertwined with other longevity regulatory mechanisms (*Savini et al., 2019*). In the long-lived *lipl-4 Tg* worms, the lysosomal enrichment of the Ragulator complex is increased with LMP-1 Lyso-IP, which may be a result of the increased proportion of mature lysosomes upon the induction of lysosomal lipolysis. At the same time, we could not rule out the possibility that the increased enrichment of the Ragulator complex is a result of the induced level of lysosomal CTNS-1/Cystinosin

in the *lipl-4 Tg* worms. We found that with LMP-1 Lyso-IP, the level of the CTNS-1/Cystinosin transporter is increased in the *lipl-4 Tg* worms, together with the increase of several cysteine cathepsins (*Figure 7G*). Our previous studies found that mitochondrial β-oxidation is increased in the *lipl-4 Tg* worms, leading to decreased triglyceride storage (*Ramachandran et al., 2019*). The *lipl-4 Tg* worms also show induced autophagy (*O'Rourke et al., 2013*; *Lapierre et al., 2011*). These phenotypes are the same as those observed in fruit flies with Cystinosin overexpression (*Jouandin et al., 2022*). Considering the inhibitory effect of Cystinosin on mTORC1 in fruit flies, the induction of Cystinosin in the *lipl-4 Tg* worms might reduce lysosomal mTORC1 signaling. Furthermore, our study reveals the lysosomal enrichment of AMPK in the *lipl-4 Tg* worms, and its requirement for the longevity effect (*Figure 5B, C*). On the other hand, the involvement of lysosomal mTORC1 and AMPK signaling in regulating the longevity effect was not identified in the *daf-2(lf)*, *isp-1(lf)*, or *glp-1(lf)* mutant. Organelle-specific signaling regulation of longevity would be interesting topics for future studies.

## Materials and methods

### *C. elegans* strains and maintenance

The following strains were used in this study: N2, CB1370 *daf-2(e1370)*, RB754 *aak-2(ok524)*, *unc-76(e911)*, MCW953 *nre-1(hd20);lin-15b(hd126)*, MCW14 *raxIs3 [ges-1p::lipl-4::SL2GFP]*, MCW859 *raxIs103[sur-5p:lmp-1::RFP-3×HA;unc-76(+)]* (*sur-5* promoter for whole-body overexpression), MCW935 *daf-2(e1370);raxIs103[sur-5p:lmp-1::RFP-3×HA;unc-76(+)]*, MCW923 *raxIs3[ges-1p::lipl-4::SL2GFP];raxIs103[sur-5p:lmp-1::RFP-3×HA;unc-76(+)]*, MCW861 *unc-76(e911);raxEx311[Pmyo-3:lmp-1::RFP-3×HA;unc-76(+)]* (*myo-3* promoter for muscle overexpression), MCW924 *unc-76(e911);raxEx346[Pcol-12:lmp-1::RFP-3×HA;unc-76(+)]* (*col-12* promoter for hypodermis overexpression), MCW862 *unc-76(e911);raxEx312[Punc-119:lmp-1::RFP-3×HA;unc-76(+)]* (*unc-119* promoter for neuron overexpression), MCW914 *unc-76(e911);raxEx341[Pges-1:lmp-1::RFP-3×HA;unc-76(+)]* (*ges-1* promoter for intestine overexpression), MCW934 *raxIs118[sur-5p:ctns-1::RFP-3×HA;unc-76(+)]*. The strains *Y58A7A.1(syb7950[Y58A7A.1::mNeonGreen])*, *R144.6(syb4893[R144.6::mNeonGreen])*, *lmp-1(syb4827[lmp-1::mNeonGreen])*, *ctns-1(syb5019[ctns-1::wrmscarlet])*, *ctns-1(syb4805[ctns-1::mNeonGreen])*, and *lmtr-3(syb8005[wrmscarlet::lmtr-3])* were generated via CRISPR/Cas9 genome editing by SunyBiotech (Fuzhou, China). The strains N2, CB1370, and RB754 were obtained from *Caenorhabditis* Genetics Center (CGC). The strain *unc-76(e911)* was obtained from Dr. Zheng Zhou's Lab. Other strains were generated in our lab.

*C. elegans* strains were maintained at 20°C on standard NGM agar plates seeded with OP50 *E. coli* (HT115 *E. coli* for RNAi experiments) using standard protocols (*Stiernagle, 2006*) and kept at least three generations without starvation before experiments.

### Molecular cloning and generating transgenics

All the expression constructs were generated using the Multisite Gateway System (Invitrogen) as previously described (*Mutlu et al., 2020*). The *lmp-1-* and *ctns-1*-coding sequences were PCR amplified from *C. elegans* cDNA then inframe fused with RFP-3×HA, and all promoters were PCR-amplified from *C. elegans* genomic DNA.

Transgenic strains were generated by microinjecting the day-1-adult germline of *unc-76(e911)* worms with DNA mixture containing expression construct and *unc-76(+)* rescuing plasmid. For integration strains, the stable extrachromosomal arrays were integrated with gamma irradiation (4500 rads for 5.9 min) and backcrossing to WT N2 at least eight times.

### Lysosome immunoprecipitation

Lyso-IP is based on the method used in mammalian cells (*Abu-Remaileh et al., 2017*). Briefly, transgenic strains stably expressing C-terminal RFP- and 3×HA-tagged lysosomal membrane protein LMP-1 or CTNS-1 under whole-body *Psur-5* or tissue-specific promoters were generated. Around 160,000 day-1-adult worms per genotype were collected, washed three times with M9 buffer then washed one time with ice-cold KPBS (Potassium (K) Phosphate Buffer Solution) (136 mM KCl, 10 mM $KH_2PO_4$). Worms in 2 ml ice-cold KPBS were quickly homogenized with Dounce homogenizer (Sigma cat. # D9063) on ice until no visible animals were seen under the microscope. The lysate was centrifuged at 1000 × *g* for 3 min at 4°C to remove debris and then the supernatant was incubated with anti-HA

magnetic beads Thermo Fisher Scientific, cat. # 88837, washed three times with ice-cold KPBS buffer before use. Each IP needs a 160 µl of beads for 6 min at 20°C with rotation. The bound beads and flow-through were separated using a magnetic stand. The bound bead fraction was washed four times with ice-cold KPBS. The bound bead and flow-through fractions were both used for LC/MS-based proteomics analyses. In order to finish processing all samples as quickly as possible, no more than three samples were processed in parallel.

## LC/MS-based proteomic analyses

The bound beads after washing were directly eluted in 100 µl of 5% sodium dodecyl sulfate (SDS) buffer and trypsin digestion was carried out using S-Trap (Protifi, NY) as per manufacturer's protocol. For the flow-through sample after IP, 100 µl sample was diluted in 5% SDS buffer and trypsin digestion was carried out using S-Trap. The peptide concentration was measured using the Pierce Quantitative Colorimetric Peptide Assay (Thermo Scientific cat. # 23275). The digested peptides were subjected to simple C18 clean-up using a C18 disk plug (3 M Empore C18) and dried in a speed vac. 1 µg of the peptide was used for LC–MS/MS analysis which was carried out using a nano-LC 1200 system (Thermo Fisher Scientific, San Jose, CA) coupled to Orbitrap Fusion Lumos ETD mass spectrometer (Thermo Fisher Scientific, San Jose, CA). The peptides were loaded on a two-column setup using a pre-column trap of 2 cm × 100 µm size (Reprosil-Pur Basic C18 1.9 µm, Dr. Maisch GmbH, Germany) and a 5 cm × 75 µm analytical column (Reprosil-Pur Basic C18 1.9 µm, Dr. Maisch GmbH, Germany) with a 75-min gradient of 5–28% acetonitrile/0.1% formic acid at a flow rate of 750 nl/min. The eluted peptides were directly electro-sprayed into a mass spectrometer operated in the data-dependent acquisition mode. The full MS scan was acquired in Orbitrap in the range of 300–1400 *m/z* at 120,000 resolution followed by top 30 MS2 in Ion Trap (AGC 5000, MaxIT 35ms, HCD 28% collision energy) with 15 s dynamic exclusion time.

The raw files were searched using the Mascot algorithm (Mascot 2.4, Matrix Science) against the *C. elegans* NCBI refseq protein database in the Proteome Discoverer (PD 2.1, Thermo Fisher) interface. The precursor mass tolerance was set to 20 ppm, fragment mass tolerance to 0.5 Da, maximum of two missed cleavage was allowed. Dynamic modification of oxidation on methionine, protein N-terminal acetylation and deamidation (N/Q) was allowed. Assigned peptides are filtered with a 1% FDR (False Discovery Rate) using Percolator validation based on *q*-value, and the Peptide Spectrum Matches output from PD2.5 will be used to group peptides onto gene levels using the 'gpGrouper' algorithm (*Saltzman et al., 2018*). This in-house program uses a universal peptide grouping logic to accurately allocate and provide MS1-based quantification across multiple gene products. Gene-protein products quantification will be performed using the label-free, intensity-based absolute quantification (iBAQ). iBAQ-based fraction of total values (iFOT) was calculated by dividing the iBAQ for each gene product by the total species iBAQ to normalize sample amount variation.

## Antibodies

Anti-*C. elegans* LMP-1, HSP-60, CYP-33, and SQV-8 monoclonal antibodies were purchased from Developmental Studies Hybridoma Bank (DSHB). Those antibodies were originally generated by Dr. Michael L. Nonet's lab (*Hadwiger et al., 2010*). Anti-β-actin antibody (C4) was purchased from Santa Cruz (sc-47778).

## Microscopy imaging
### Regular microscopy

Tissue-specific lyso-tag expression example images (*Figure 3A*) were captured using Leica DMi8 THUNDER Imaging Systems using ×20 objective. The images that show the colocalization between CRISPR knock-in lines LMP-1::mNeonGreen and CTNS-1::wrmScarlet were captured by Zeiss LSM 980 with Airyscan. The images that show the colocalization between wrmScarlet::LMTR-3 and LMP-1::mNeonGreen/CTNS-1::mNeonGreen were taken using Nikon CSU-W1 spinning disk confocal microscopy system. Other microscopy images were captured using an Olympus FV3000 confocal microscopy system using ×60 or ×20 objective. *C. elegans* were anesthetized in 1% sodium azide in M9 buffer and placed on a 2% agarose pad sandwiched between the glass microscopic slide and coverslip.

## Fluorescence lifetime microscopy

L1 RNAi sensitive *nre-1(hd20);lin-15b(hd126)* worms were seeded on 3.5 cm RNAi plates and raised at 20°C for 2 days, and then around 20 worms each well were transferred to the 3.5 cm RNAi plates containing RNAi bacteria and 0.5 μM of LysoSensor Green DND-189 (Invitrogen L7535) and raised for 18 hr (in dark) at 20°C. The worms were imaged using ISS Q2 Time-resolved Laser Scanning Confocal Nanoscope. The laser excitation wavelength was set at 476 nm and the 500–633 nm emission filter was used to detect the LysoSenor Green signal. The first pair of intestinal cells of each worm was imaged. The lifetimes of LysoSensor-containing puncta were measured using ISS VistaVision software. The pH of the lysosome was then calculated based on the LysoSensor lifetime-pH calibration curve (*Deng et al., 2023*).

## LysoSensor RNAi screen

The primary screen was performed on 95 lysosomal-enriched candidates shared between the LMP-1 and the CTNS-1 Lyso-IP proteomic profiling datasets. Each RNAi bacteria clone was seeded onto 12-well RNAi plates containing 1 mM IPTG (Isopropyl β-D-1-thiogalactopyranoside) and allowed to dry. The dried plates were then incubated at room temperature overnight to induce dsRNA expression. Synchronized L1 *nre-1(hd20);lin-15b(hd126)* worms were seeded on 12-well RNAi plates and raised at 20°C for 2 days, and then around 30 worms each well were transferred to the RNAi plates containing RNAi bacteria and 0.5 μM of LysoSensor Green DND-189. After 18 hr, LysoSensor signals were examined by the naked eyes using a Nikon SMZ18 fluorescence stereo microscope. The candidates with obvious LysoSensor alteration were selected for the secondary LysosSensor RNAi screen. In the secondary screen, worms stained using LysoSensor Green DND-189 were imaged by the Olympus FV3000 confocal microscopy system. The changes in LysoSensor signals in the first pair of intestine cells were quantified by ImageJ (including intensity, size, and number).

## Lysosome distribution quantification

The quantification method was modified from previous publications on lysosomal distribution in mammalian cell lines (*Johnson et al., 2016*; *Willett et al., 2017*). Images were first captured using Olympus Fluoview software and imported into Matlab by the Bio-Formats tool (*Linkert et al., 2010*). Next, the cell membrane and nuclear membrane are outlined manually. The algorithm (code included in Supplementary materials) determines the geometric center of the nucleus and radiates at all angles to locate line segments between the nuclear and cell membrane. The line segments are then evenly divided and circled to segment the cytosol into different regions. We calculate: (1) The mean RFP fluorescence distribution across regions (normalized to the mean intensity of the whole cell to avoid variations of RFP expression across cells) (*Figure 6D and F*); (2) The cumulative intensity distribution from the most perinuclear region to the most peripheral region (normalized to the overall intensity of the whole cell) (*Figure 6—figure supplement 1B, C*). The algorithm requires a convex shape of the cell, and most of the gut cells imaged meet this need.

## Lifespan assays

Worms were synchronized by bleach-based egg preparation and subsequent starvation in M9 buffer for over 24 hr. Synchronized L1 worms were placed on the plates, and animals were synchronized again by manual picking at mid L4 stage and marked as day 0. Approximately 90–120 worms were placed in three parallel plates for each condition. Worms are hence observed and transferred to freshly made RNAi plates every other day. Animals are categorized as alive, dead (cessation of movement in response to platinum wire probing), or censored. The statistical analyses were performed with the SPSS23 Kaplan–Meier survival function and the log-rank test. GraphPad Prism 9 was used to graph the results.

## Structural stimulation by AlphaFold2

The structure of R144.6 (UniProt: Q10000) was predicted by AlphaFold2 and downloaded from The AlphaFold Protein Structure Database (https://alphafold.ebi.ac.uk/) (*Jumper et al., 2021*; *Varadi et al., 2022*). The molecular graphics of the R144.6 structure were performed with UCSF Chimera (*Pettersen et al., 2004*).

## Statistical methods

PCA was performed by R package Factextra (*Le et al., 2008*) with the normalized iFOT abundance of proteins detected as the input. The Pearson correlation matrices and coefficient (*r*) among replicates were generated by GraphPad Prism 9. The unpaired, two-tailed *t*-tests for the multiple comparisons were used to calculate the p values in the correlation analyses by GraphPad Prism 9.

## Acknowledgements

We thank A Dervisefendic and P Svay for maintenance support; Mass Spectrometry Proteomics Core of Baylor College of Medicine for mass spectrometry proteomic analysis; Z Zhou (Baylor College of Medicine) for sharing *unc-76(e911)* strain. Some strains were obtained from the *Caenorhabditis* Genetics Center (CGC), which is funded by the NIH Office of Research Infrastructure Programs (P40 OD010440). MCW is currently supported by Howard Hughes Medical Institute, and YY is currently supported by the National Natural Science Foundation of China (32071146).

## Additional information

### Funding

| Funder | Grant reference number | Author |
|---|---|---|
| Howard Hughes Medical Institute | | Meng C Wang |
| National Natural Science Foundation of China | 32071146 | Yong Yu |

The funders had no role in study design, data collection, and interpretation, or the decision to submit the work for publication.

### Author contributions

Yong Yu, Conceptualization, Resources, Data curation, Formal analysis, Supervision, Funding acquisition, Validation, Investigation, Visualization, Methodology, Writing – original draft, Project administration, Writing – review and editing; Shihong M Gao, Conceptualization, Resources, Data curation, Software, Formal analysis, Validation, Investigation, Visualization, Methodology, Writing – original draft, Writing – review and editing; Youchen Guan, Conceptualization, Resources, Data curation, Formal analysis, Validation, Investigation, Visualization, Methodology, Writing – original draft, Writing – review and editing; Pei-Wen Hu, Data curation, Validation, Visualization, Methodology; Qinghao Zhang, Data curation, Formal analysis, Writing – review and editing; Jiaming Liu, Bentian Jing, Data curation, Formal analysis; Qian Zhao, Formal analysis, Visualization; David M Sabatini, Methodology; Monther Abu-Remaileh, Methodology, Writing – review and editing; Sung Yun Jung, Data curation, Supervision; Meng C Wang, Conceptualization, Data curation, Formal analysis, Supervision, Funding acquisition, Methodology, Writing – original draft, Project administration, Writing – review and editing

### Author ORCIDs

Yong Yu ⓘ https://orcid.org/0000-0002-9821-9726
Shihong M Gao ⓘ http://orcid.org/0000-0003-3238-8348
Youchen Guan ⓘ http://orcid.org/0000-0002-6570-5293
Meng C Wang ⓘ https://orcid.org/0000-0002-5898-6007

### Decision letter and Author response

Decision letter https://doi.org/10.7554/eLife.85214.sa1
Author response https://doi.org/10.7554/eLife.85214.sa2

## Additional files

### Supplementary files

• Supplementary file 1. Lysosome-enriched proteins identified from LMP-1 Lyso-IP using wild-type

(WT) worms.

- Supplementary file 2. Lysosome-enriched proteome exhibits tissue specificity.
- Supplementary file 3. Lysosome-enriched proteins identified from LMP-1 Lyso-IP using *lipl-4 Tg* worms.
- Supplementary file 4. Lysosome-enriched proteins identified from LMP-1 Lyso-IP using *daf-2(lf)* mutant.
- Supplementary file 5. Lysosome-enriched proteins identified from LMP-1 Lyso-IP using *isp-1(lf)* mutant.
- Supplementary file 6. Lysosome-enriched proteins identified from LMP-1 Lyso-IP using *glp-1(lf)* mutant in 25°C.
- Supplementary file 7. Lysosome-enriched proteins identified from LMP-1 Lyso-IP using wild-type (WT) worms in 25°C.
- Supplementary file 8. Summary of lifespan analyses.
- Supplementary file 9. Lysosome-enriched proteins identified from CTNS-1 Lyso-IP using wild-type (WT) worms.
- Supplementary file 10. LysoSensor screening of lysosome-enriched proteins shared between LMP-1 and the CTNS-1 Lyso-Ips.
- Supplementary file 11. File list of mass spectrum samples.
- MDAR checklist
- Source code 1. Matlab code for lysosome distribution quantification.

## Data availability

The mass spectrometry data for protein identification have been deposited via the MASSIVE repository (MSV000093375) to the ProteomeXchange Consortium with the dataset identifier PXD046869 (http://proteomecentral.proteomexchange.org/cgi/GetDataset?ID=PXD046869). Analysis code for Figure 6D, F, and Figure 6-figure supplement 1 is included in *Source code 1*.

The following dataset was generated:

| Author(s) | Year | Dataset title | Dataset URL | Database and Identifier |
|---|---|---|---|---|
| Yu Y, Gao SM, Guan Y, Pw Hu, Zhang Q, Liu J, Jing B, Zhao Q, Sabatini DM, Abu-Remaileh M, Jung SY, Wang MC | 2024 | Organelle proteomic profiling reveals lysosomal heterogeneity in association with longevity | http://proteomecentral.proteomexchange.org/cgi/GetDataset?ID=PXD046869 | ProteomeXchange, PXD046869 |

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
