## [Editor Report]

The authors present a powerful tool to unbiasedly identify lysosome-associated proteins in *C. elegans*, and they provide a compelling, in-depth assessment of how this method can be used to understand longevity pathways and identify novel proteins. Understanding lysosomal differences in specific tissues or in response to different longevity conditions are exciting as it provides new insight into how organelles could control specific homeostasis responses. This valuable tool and proteomics datasets also represent a great resource for the *C. elegans* community and should pry open new studies on the regulation and role of the lysosome at the organismal level.

---

## [Decision Letter]

**Decision letter after peer review:**

Thank you for submitting your article "Organelle proteomic profiling reveals lysosomal heterogeneity in association with longevity" for consideration by *eLife*. Your article has been reviewed by 3 peer reviewers, including Kang Shen as Reviewing Editor and Reviewer #1, and the evaluation has been overseen by Suzanne Pfeffer as the Senior Editor.

Essential revisions:

1) Please consider the technical comments from the reviewers when you revise your manuscript.

2) Two out of three reviewers commented on the notion that lipl-4 nuclear signaling is not fully demonstrated. You can decide if you want to tamper the conclusions or provide more experiments to strengthen the argument.

*Reviewer #1 (Recommendations for the authors):*

1. It should be made clear that the lmp-1 transgene is an overexpressed transgene. Please specify which promoter the fusion is expressed under and does that achieve expression in all cell types.

2. In Figure 3, among the three samples of the daf-2 proteomics, IP-1 looks similar to the Lipl-4 Tg profile, which would make sense because both strains are long-lived. However, the other two samples of the daf-2 worms do not look like IP-1. Is there an explanation for that?

3. In Figure 3, is there a significant difference in life span between wt and aak-2 mutants? If there is not any difference, then I agree that the data strongly suggest that aak-2 is responsible for the lipl-4 overexpression-induced increase in life span. If the aak-2 mutant is short-lived on its own, the argument is a bit tricky because they might function in parallel pathways.

4. The genetics experiments presented in Figure 4 are compelling. It seems to me that the lysosomes become more clustered in the lipl-4 Tg worms. Only a small fraction of these "clustered" lysosomes are making physical contact with the nuclear envelope. I am not sure if it is necessary to propose that lysosomal nuclear contact is necessary for this signaling to occur. Overall, this figure is very nice.

5. Why knockdown of unc-32 decrease the pH in lysosomes? Shouldn't it be the other way around?

*Reviewer #2 (Recommendations for the authors):*

1) For the tissue specificity, it would strengthen the study if the lysosomal localization could be confirmed for selected hits (similar to what was done for R144.6 in 7J). Another alternative could be to analyze in which tissues lysosomal proteins from long-lived worms are localized.

2) Could the authors comment if the presence of the IP-tag does affect health/development?

3) Could the authors provide a more detailed section (e.g. in the methods) on the analysis of the Lyso-IP after mass spectrometry? While part of this information can be found in previous publications (e.g. Abu-Remaileh et a. 2017), it would be helpful to have a method paragraph and a diagram for the analysis given the different inputs (tissue-specific vs high lysosome abundance in the lipl-4 Tg). For example: is the data normalized to whole protein/amount of protein pulled down? Would having more lysosomes (e.g. in lipl-4) increase all lysosome-associated proteins? Generally, what is the amount of protein that can be eluted from the beads after the IP?

4) Could the authors provide more details in the method section on the Lyso-IP itself (e.g. temperature of some buffers, type of Dounce homogenizer, how many samples are processed in parallel, etc)?

5) Overall, the study nicely shows the use of the Lyso-IP technique for discoveries in different areas (different longevity mutants, different tissues, different steps of lysosome maturation). However, because of the breadth covered by this manuscript, the study also feels a bit disconnected. It could be helpful to focus the manuscript a bit more on the key biological findings.

*Reviewer #3 (Recommendations for the authors):*

1. Using proteomics authors show that LMP1- lysosomes are partially distinct from the CTNS1-expressing ones. However, these two proteins are thought to generally co-localize in most late endosomes/lysosomes. Co-localization between the two proteins should be shown across different worm tissues to determine whether the two vesicle populations are truly distinct, or whether differences in pull-down efficiency could explain the results.

2. Similarly, the identification of the LAMTOR/SLC38A9 proteins in the CTNS but not LMP lysosomes is puzzling, given the extensive colocalization between mTORC1 proteins and LAMP1/2 reported in the literature. Their colocalization, or lack thereof, in worms should be documented via direct imaging.

3. The significance of the perinuclear positioning of Lipl4-Tg worms needs to be further clarified. Clustering near the nucleus does not imply direct interaction between lysosomal and nuclear membranes. EM analysis should be performed to clarify whether direct contacts indeed exist.

4. Furthermore, the KD of Npp-6 seems arbitrary, as there is no evidence that this protein may serve as a bridge between lysosomes and the nuclear envelope. Moreover, its loss may affect longevity by disrupting nuclear import-export. In the absence of an in-depth analysis of this process, the conclusion that 'enhanced lysosome-nucleus interaction is specifically related to LIPL-4-induced lysosomal signaling and longevity should be removed.

---

## [Author Response]

Essential revisions:1) Please consider the technical comments from the reviewers when you revise your manuscript.

As suggested by the reviewers, we have included additional technical details in the Materials and Methods section.

2) Two out of three reviewers commented on the notion that lipl-4 nuclear signaling is not fully demonstrated. You can decide if you want to tamper the conclusions or provide more experiments to strengthen the argument.

We appreciate the reviewers’ comments. We agree with the reviewers that our data did not support increased lysosome-nucleus contacts and the result from light microscopy analysis only revealed the increased distribution of lysosomes around the nucleus. As suggested by the reviewers, the *npp-6* RNAi result may be related to nuclear import/export. We have thus conducted new experiments to knockdown *ima3* and *xpo-1,* the *C. elegans* homologs of importin α and exportin, respectively. We found that the inactivation of *ima-3* but not *xpo-1* suppresses the lifespan extension caused by *lipl-4 Tg*, suggesting the importance of nuclear import. We have added these new results and revised the manuscript to tamper the conclusion on lysosome-nucleus contacts.

Reviewer #1 (Recommendations for the authors):1. It should be made clear that the lmp-1 transgene is an overexpressed transgene. Please specify which promoter the fusion is expressed under and does that achieve expression in all cell types.

We thank the reviewer for the suggestion. In the revised manuscript, we have specified that the *lmp-1* transgene is an overexpressed transgene. We have also specified that the whole-body expression is driven by the *sur-5* promoter, and the intestine-, muscle-, neuron-, and hypodermis-specific expression are driven by the *ges^-1^*, *myo-3*, *unc-119*, and *col-12* promoters, respectively.

2. In Figure 3, among the three samples of the daf-2 proteomics, IP-1 looks similar to the Lipl-4 Tg profile, which would make sense because both strains are long-lived. However, the other two samples of the daf-2 worms do not look like IP-1. Is there an explanation for that?

We thank the reviewer for raising this point. We have thoroughly examined the raw data and have confirmed that all the samples were prepared using the same method. Therefore, we believe that the observed variation underscores the importance of including multiple replicates in this type of proteomic analysis, and we must include all the replicates to avoid biased conclusions.

We appreciate the reviewer’s idea that long-lived strains may share similar lysosome-enriched proteomes and thus conducted new proteomic analyses on two long-lived strains: the *isp-1* mutant that reduces mitochondrial electron transport chain complex III activity, and the *glp-1* mutant that has a defective germline. The new results show that neither the *isp-1* mutant nor the *glp-1* mutant exhibits the same lysosome-enriched proteome changes as seen in the *lipl-4 Tg* strain.

3. In Figure 3, is there a significant difference in life span between wt and aak-2 mutants? If there is not any difference, then I agree that the data strongly suggest that aak-2 is responsible for the lipl-4 overexpression-induced increase in life span. If the aak-2 mutant is short-lived on its own, the argument is a bit tricky because they might function in parallel pathways.

As shown in the Supplementary Table 7, the lifespan of the *aak-2* mutant worms with *aak-1* RNAi is significantly shorter than that of wild type worms, and reduction is 18%. In the *lipl-4 Tg* background, the lifespan reduction caused by the *aak-2* mutant and *aak-1* RNAi is 29%. Therefore, we concluded that *aak-1* and *aak-2* are partially responsible for the lifespan extension caused by the *lipl-4* overexpression. We have added the statistical comparison between the *aak-2* mutant worms with *aak1* RNAi and wild type worms into Figure 5C and revised the manuscript to further increase the textural clarity of the conclusion.

4. The genetics experiments presented in Figure 4 are compelling. It seems to me that the lysosomes become more clustered in the lipl-4 Tg worms. Only a small fraction of these "clustered" lysosomes are making physical contact with the nuclear envelope. I am not sure if it is necessary to propose that lysosomal nuclear contact is necessary for this signaling to occur. Overall, this figure is very nice.

We thank the reviewer for her/his comment and suggestion. We agree with the reviewer that the current result supports the increased peri-nuclear clustering of lysosomes, but not necessarily increased lysosome-nucleus contact. We have also conducted new experiments to knockdown *ima-3* and *xpo-1,* the *C. elegans* homologs of importin α and exportin, respectively. We found that the inactivation of *ima-3* but not *xpo-1* suppresses the lifespan extension caused by *lipl-4 Tg*, suggesting the importance of nuclear import. We have added these new results and revised the manuscript to tamper with the conclusion on the lysosome-nucleus contact.

5. Why knockdown of unc-32 decrease the pH in lysosomes? Shouldn't it be the other way around?

We agree with the reviewer that this result seems to be contradictory to what we would expect. We speculate that this pH decrease is to compensate the loss of acidic lysosomes. In the analysis shown in the manuscript, we used LysoSensor Green to label lysosomes. This dye becomes fluorescent in the acidic environment, and its fluorescence lifetime is negatively correlated with pH (PMID: 11444806, doi: https://doi.org/10.1101/2023.09.25.559395). As shown in Figure 8E and 8F, the number of acidic lysosomes that are LysoSensor Green positive is largely decreased (5-fold) with *unc-32* RNAi knockdown. To compensate for this big loss, the remaining lysosomes may become more acidic. But overall, there are less acidic lysosomes with the *unc-32* RNAi knockdown. We thank the reviewer for raising this point.

Reviewer #2 (Recommendations for the authors):1) For the tissue specificity, it would strengthen the study if the lysosomal localization could be confirmed for selected hits (similar to what was done for R144.6 in 7J). Another alternative could be to analyze in which tissues lysosomal proteins from long-lived worms are localized.

We thank the reviewer’s comments. As suggested by the reviewer, we have also generated a CRISPR knock-in line for one hypodermis-specific candidate Y58A7A.1 that encodes a copper transporter, and validated its hypodermis-specific lysosomal localization (new Supplementary Figure 2E).

2) Could the authors comment if the presence of the IP-tag does affect health/development?

We appreciate the reviewer’s helpful suggestion. We did not observe that the presence of the IP-tag affects worms’ health/development. In the revised manuscript, we have included new results showing the developmental timing and lifespan of the transgenic strains expressing the IP-tag (new Supplementary Figure 1A and 1B).

3) Could the authors provide a more detailed section (e.g. in the methods) on the analysis of the Lyso-IP after mass spectrometry? While part of this information can be found in previous publications (e.g. Abu-Remaileh et a. 2017), it would be helpful to have a method paragraph and a diagram for the analysis given the different inputs (tissue-specific vs high lysosome abundance in the lipl-4 Tg). For example: is the data normalized to whole protein/amount of protein pulled down? Would having more lysosomes (e.g. in lipl-4) increase all lysosome-associated proteins? Generally, what is the amount of protein that can be eluted from the beads after the IP?

As suggested by the reviewer, we have extended the method section on Lyso-IP to include more details. We believe that the new version should be sufficient for any lab to follow this protocol and conduct their own analyses. We will also take advantage of the *eLife* “Request a Protocol” feature to share the detailed version of the Lyso-IP method with researchers who are interested.

“Is the data normalized to whole protein/amount of protein pulled down?” The method of quantification of purified proteins is well described one of the references (Saltzman et al., 2018). We have included a brief description in the method: Assigned peptides are filtered with a 1% FDR using Percolator validation based on q-value, and the Peptide Spectrum Matches (PSMs) output from PD2.5 will be used to group peptides onto gene levels using ‘gpGrouper’ algorithm (Saltzman et al., 2018). This in-house program uses a universal peptide grouping logic to accurately allocate and provide MS1 based quantification across multiple gene products. Gene-protein products (GPs) quantification will be performed using the label-free, intensity-based absolute quantification (iBAQ). iBAQ-based fraction of total values (iFOT) was calculated by dividing the iBAQ for each gene product by the total species iBAQ to normalize sample amount variation.

“Would having more lysosomes (e.g. in lipl-4) increase all lysosome-associated proteins?” We believe that this should not be the case, considering that the same amount of proteins from Lyso-IP was loaded into the MS and therefore more lysosomes should not simply lead to a higher enrichment.

“Generally, what is the amount of protein that can be eluted from the beads after the IP?” As now explained in the method section, we did not elute proteins from the beads after the IP; instead, we conducted trypsin digestion directly. For each Lyso-IP and Flowthrough sample, 1µg of peptides was loaded into the MS.

5) Overall, the study nicely shows the use of the Lyso-IP technique for discoveries in different areas (different longevity mutants, different tissues, different steps of lysosome maturation). However, because of the breadth covered by this manuscript, the study also feels a bit disconnected. It could be helpful to focus the manuscript a bit more on the key biological findings.

We thank the reviewer for raising this point. Although the primary goal of this manuscript is to provide new method and resource to the community, we agree with the reviewer that it would be helpful to highlight key biological findings. We have revised the manuscript thoroughly aiming to balance the focus between method/data sharing and new discovery reporting.

Reviewer #3 (Recommendations for the authors):1. Using proteomics authors show that LMP1- lysosomes are partially distinct from the CTNS1-expressing ones. However, these two proteins are thought to generally co-localize in most late endosomes/lysosomes. Co-localization between the two proteins should be shown across different worm tissues to determine whether the two vesicle populations are truly distinct, or whether differences in pull-down efficiency could explain the results.

We thank the reviewer for the comment. We have generated CRISPR knock-in lines for tagging endogenous CTNS-1 with wrmScarlet and LMP-1 with mNeonGreen. Using these lines, we have examined the co-localization between CTNS-1 and LMP-1 in the intestine, hypodermis, muscle, and neurons. We found that CTNS-1 and LMP-1 positive vesicles are only partially overlapped (new Supplementary Figure 5), which is consistent with the proteomic results.

2. Similarly, the identification of the LAMTOR/SLC38A9 proteins in the CTNS but not LMP lysosomes is puzzling, given the extensive colocalization between mTORC1 proteins and LAMP1/2 reported in the literature. Their colocalization, or lack thereof, in worms should be documented via direct imaging.

We appreciate the reviewer’s comment. We agree that our statement that “the Ragulator complex (LMTR-2, 3 and 5) is enriched at the lysosome purified from WT worms using CTNS-1 but not LMP-1 Lyso-IP” is misleading. As shown in new Figure 7E, LMTR-2, 3 and 5 were identified in both the CTNS-1 (three out of three replicates) and LMP-1 lysosome (three out of four replicates for LMTR-2, two out of four replicates for LMTR-3, three out of four replicates for LMTR-5), and the enrichment is higher in the CTNS-1 lysosome than in the LMP-1 lysosome. In the revision, we have updated the statement to “the Ragulator complex (LMTR-2, 3 and 5) exhibited a higher enrichment in the lysosome purified from WT worms using CTNS-1 Lyso-IP than using LMP-1 Lyso-IP”. Furthermore, we have generated a CRISPR knock-in line for tagging endogenous LMTR-3 with mNeonGreen and crossed this line with the CRISPR knock-in lines in which LMP-1 or CTNS-1 is tagged with wrmScarlet. We found that LMTR-3 shows a complete overlap with CTNS-1 in the intestine, muscle, and hypodermis (new Figure 7F, new Supplementary 6C), but it only shows a complete overlap with LMP-1 in the intestine, not in the muscle or hypodermis (new Figure 7F, new Supplementary 6D).

3. The significance of the perinuclear positioning of Lipl4-Tg worms needs to be further clarified. Clustering near the nucleus does not imply direct interaction between lysosomal and nuclear membranes. EM analysis should be performed to clarify whether direct contacts indeed exist.

We thank the reviewer for the comment. We agree with the reviewer that the current result supports the increased peri-nuclear clustering of lysosomes, but not necessarily increased direct lysosome-nucleus membrane contact. As suggested by the reviewers and editor, we have revised the manuscript to avoid the overstatement on the lysosome-nucleus contact.

4. Furthermore, the KD of Npp-6 seems arbitrary, as there is no evidence that this protein may serve as a bridge between lysosomes and the nuclear envelope. Moreover, its loss may affect longevity by disrupting nuclear import-export. In the absence of an in-depth analysis of this process, the conclusion that 'enhanced lysosome-nucleus interaction is specifically related to LIPL-4-induced lysosomal signaling and longevity should be removed.

We thank the reviewer for raising the interesting point on nuclear import-export. We have conducted new experiments to knockdown *ima-3* and *xpo-1,* the *C. elegans* homologs of importin α and exportin, respectively. We found that the inactivation of *ima-3* but not *xpo-1* suppresses the lifespan extension caused by *lipl-4 Tg*, suggesting the importance of nuclear import. In addition, we have conducted new proteomic analyses in the long-lived *isp-1* mutant that reduces mitochondrial electron transport chain complex III activity. We found that NPP-6 is not enriched in the lysosome purified from the *isp-1* mutant. Consistently, *npp-6* RNAi knockdown did not suppress the lifespan extension in the *isp-1* mutant. We have included these new results in the revised manuscript and updated the conclusion to:

“These results suggest that the nucleoporin protein NPP-6 is specifically involved in the regulation of lysosomal LIPL-4-induced longevity.”